# Quantifying the impact of synoptic circulation patterns on ozone variability in North China from April-October 2013-2017

Jingda Liu[1,2], Lili Wang[2,4], Mingge Li[2,5], Zhiheng Liao[6], Yang Sun[2], Tao Song[2], Wenkang Gao[2], Yonghong Wang[4], Yan Li[7], Dongsheng Ji[2], Bo Hu[2], Veli-Matti Kerminen[4], Yuesi Wang[1,2,3,5], Markku Kulmala[4]

[1]Department of Atmospheric Physics, Nanjing University of Information Science & Technology, Nanjing 210044, China

[2]State Key Laboratory of Atmospheric Boundary Layer Physics and Atmospheric Chemistry (LAPC), Institute of Atmospheric Physics, Chinese Academy of Sciences, Beijing 100029, China

[3]Centre for Excellence in Atmospheric Urban Environment, Institute of Urban Environment, Chinese Academy of Science, Xiamen, Fujian 361021, China

[4]Institute for Atmospheric and Earth System Research / Physics, Faculty of Science, University of Helsinki, Finland

[5]University of Chinese Academy of Sciences, Beijing 100049, China

[6]School of Atmospheric Sciences, Sun Yat-sen University, Guangzhou, Guangdong, China

[7]Fangshan Meteorological Bureau, Beijing, 102488, China

*Correspondence to:* Lili Wang (wll@mail.iap.ac.cn)

## Abstract

The characteristics of ozone variations and the impacts of synoptic and local meteorological factors in North China were quantitively analyzed during the warm season from 2013 to 2017 based on multicity in situ ozone and meteorological data as well as meteorological reanalysis. The domain-averaged maximum daily 8-h running average $O_3$ (MDA8 $O_3$) concentration was $122\pm11$ µg m$^{-3}$, with an increase rate of 7.88 µg m$^{-3}$ year$^{-1}$, and the three most polluted months were closely related to the variations in the synoptic circulation patterns, which occurred in June (149 µg m$^{-3}$), May (138 µg m$^{-3}$) and July (132 µg m$^{-3}$). Twenty-six weather types (merged into 5 weather categories) were objectively identified using the Lamb-Jenkinson method. The highly-polluted weather categories included S-W-N directions (geostrophic wind direction diverts from south to north), low-pressure related weather types (LP) and cyclone type, which the study area controlled by low-pressure center (C), and the corresponding domain-averaged MDA8 $O_3$ concentrations were 122, 126 and 128 µg m$^{-3}$, respectively. Based on the frequency and intensity changes of the synoptic circulation patterns, 39.2% of the interannual increase in the domain-averaged $O_3$ from 2013 to 2017 was attributed to synoptic changes, and the intensity of the synoptic circulation patterns was the dominant factor. Using synoptic classification and local meteorological factors, the segmented synoptic-regression approach was established to evaluate and forecasted daily ozone variability on an urban scale. The results showed that this method is practical in most cities, and the dominant factors are the maximum temperature, southerly winds, relative humidity on the previous day and on the same day, and total cloud cover. Overall, 41-63% of the day-to-day variability in the MDA8 $O_3$ concentrations was due to local meteorological variations in most cities over North China, except for two cities: QHD (Qinhuangdao) at 34% and ZZ (Zhengzhou) at 20%. Our

quantitative exploration of the influence of both synoptic and local meteorological factors on interannual
and day-to-day ozone variability will provide a scientific basis for evaluating emission reduction
measures that have been implemented by the national and local governments to mitigate air pollution in
North China.

## 1 Introduction

Tropospheric ozone ($O_3$) is one of the air pollutants of greatest concern due to its considerable harm
to human health and vegetation (Kinney, 2008; Fleming et al., 2018; Mills et al., 2018). $O_3$ is formed
through nonlinear interactions between $NO_x$ and volatile organic compounds in combination with
sunlight (Monks et al., 2009; Monks et al., 2015). Thus, ozone levels are controlled by precursors and
meteorological conditions. With industrialization advancement and rapid economic growth, North China
has become one of the most populated and polluted regions in the world. The national and local
governments have implemented a series of measures to reduce emissions since 2013, and although $PM_{2.5}$
has decreased significantly, $O_3$ pollution is still severe in this region (Lu et al., 2018; Li et al., 2019).
Several studies have explored the variation in summer ozone in China (He et al., 2017; Liao et al., 2017;
Lu et al., 2018; Li et al., 2019). However, systematic research aimed at quantifying the evolution of
ozone and meteorological impacts and contributions throughout the warm season (April-October) was
limited during the five years (2013-2017) when the Action Plan for Air Pollution Prevention and Control
(www.gov.cn/zwgk/2013-09/12/content_2486773.htm) was implemented. This lack of analysis has
prevented a clear understanding of the effect of emission reduction measures on ozone in North China
from being obtained.
Meteorological factors affect ozone levels through a series of complex combinations of processes,
including emissions, transport, chemical transformations and removal (Chan and Yao, 2008; Jacob and
Winner, 2009; Lu et al., 2019). Meteorological conditions are the primary factor that determine the day-
to-day variations in pollutant concentrations over China (He et al., 2016; He et al., 2017), whereas long-
term $O_3$ trends are influenced both by climatological (weather types, temperature, humidity, and radiation,
etc.) and environmental factors (changes in anthropogenic and natural sources). Therefore, the impact of
reduced anthropogenic emissions on $O_3$ variations can be estimated more accurately if we are able to
quantify the meteorological influence.
Synoptic meteorological conditions have an important effect on regional ozone distribution and
variation (Shen et al., 2015). A given synoptic circulation pattern represents a particular range of
meteorological conditions; therefore, synoptic classification is a useful method for gaining insight into
the impact of meteorology on ozone levels at regional scale. Previous studies have demonstrated a
significant connection between the weather type and surface $O_3$ concentration; however, the relation
between these two quantities varies in different regions due to differences in the topography, pollution
source, local circulation, etc.(Moody et al., 1998; Cooper et al., 2001; Hegarty et al., 2007; Demuzere et
al., 2009; Monks et al., 2009; Wang et al., 2009a; Zhang et al., 2012; Zhang et al., 2013; Pope et al.,
2016; Liao et al., 2017). For example, based on the Lamb-Jenkinson weather typing technique, Demuzere
et al. (2009) demonstrated increased surface $O_3$ concentrations in summer in an easterly weather type at
a rural site in Cabauw, Netherlands, whereas the opposite result was obtained by Liao et al. (2017) in the
Yangtze River Delta region in eastern China. Therefore, synoptic classification and its relationship with
$O_3$ need to be explored separately in different regions. In addition, based on synoptic classification,
Comrie and Yarnal (1992) and Hegarty et al. (2007) suggested a reconstructed pollutant concentration
(caused by synoptic influence) algorithm, which can separate the climatological and environmental
variability in environmental data. It was found that 46% and 50% of the interannual variability in the $O_3$
concentration was reproduced in the northeastern United States (Hegarty et al., 2007) and Hong Kong
(Zhang et al., 2013), respectively, by taking into account the interannual changes in the frequency and
intensity of synoptic patterns.
At the urban scale, the daily variation in the ozone concentration is affected by both synoptic and local
meteorological factors. Quantifying the contribution of local meteorological factors to day-to-day
variations in ozone concentrations will provide a scientific basis and guidance for reasonable ozone
reduction measures, and clarifying and quantifying the relationship between meteorological factors and
ozone concentration is vital for daily forecasts of ozone pollution potential. Weather type classification
prior to regression analysis is superior to a simple linear regression approach (Eder et al., 1994; Barrero
et al., 2006; Demuzere et al., 2009; Demuzere and van Lipzig, 2010), and synoptic-regression-based
algorithm can reproduce the observed $O_3$ distributions and provide a better parameterization to promote
the understanding of the dependence of ozone on meteorological factors in a given urban region.
Overall, in this study, we explore how the maximum daily 8-h running average $O_3$ (MDA8 $O_3$)
concentration varies and quantify the contributions of synoptic and local meteorological conditions to
the ozone variability in North China (58 cities covering Hebei, Shanxi, Shandong, and Henan Provinces
and Beijing and Tianjin municipalities) during April-October in 2013-2017. Our specific goals are to 1)
demonstrate the characteristics and variation trends in the surface MDA8 $O_3$ concentration; 2) classify
the predominant weather types and meteorological mechanisms underlying the regional ozone levels and
variability; 3) quantify the contributions of changes in synoptic circulation patterns (frequency and
intensity) to the interannual variability in the $O_3$ concentration; and 4) quantify the contributions of local
meteorological factors to day-to-day variations in $O_3$ levels, and identify the prominent meteorological
variables and construct $O_3$ potential forecast model for major cities..
**2 Data and methods**
**2.1 Ozone and PM$_{2.5}$ data**
The hourly $O_3$ and PM$_{2.5}$ data during April-October, 2013-2017 were derived from the National Urban
Air Quality Real-time Publishing Platform (http://106.37.208.233:20035/). According to technical
regulation for ambient air quality assessment (HJ 663‐2013, http://www.mee.gov.cn/), the MDA8 $O_3$
concentration was calculated for each monitoring site based on the hourly data from the time period
08:00-24:00 for the days with at least 14 hours of measurement data. If less than 14 hours of valid data
are available, the results are still valid if the MDA8 $O_3$ concentration exceeds the national
concentration limit standard. Each city has at least two monitoring sites, and the MDA8 $O_3$ levels for a
city are the corresponding averages over all sites in that city. The MDA8 $O_3$ values were collected in
only 14 cities for the time period from 2013-2017 and in an additional 44 cities for the time period
from 2015-2017, and the detailed information is shown in Fig. 1 and Table S1. The original units for
the ozone observations was μg m$^{-3}$, and the conversion coefficient from the mixing ratios (unit: ppbv)
to μg m$^{-3}$ was a constant (e.g., 0.5 at a temperature of 25 °C and pressure of 1013.25 hPa). In this study,
we used the original units. Unless otherwise noted, the analysis of $O_3$ refers to MDA8 $O_3$ during April-
October in this paper.

**122**  **2.2 Meteorological data**

**123**  Gridded-mean sea level pressure data, 10-meter U and V wind components ($U_{10}$ and $V_{10}$, respectively),

**124**  boundary layer height (BLH) and 2-meter temperature ($T_2$) with a 1° horizontal resolution and vertical

**125**  velocity (ω) from 1000-100 hPa (27 levels) and wind divergence (div) from 1000-850 hPa (7 levels) in

**126**  6-h intervals (Beijing time 02, 08, 14 and 20) for 2013-2017 were obtained from the European Center

**127**  for Medium Weather Forecast Reanalysis Interim (ERA-Interim).

**128**     Four measurements per day for temperature (T), relative humidity (RH), total cloud cover (TCC), rain,

**129**  wind speed (ws), wind direction (wd), and pressure (pre) in 58 cities during April-October 2013-2017

**130**  were obtained from the China Meteorological Administration in the Meteorological Information

**131**  Combine Analysis and Process System (MICAPS). Then, daily mean meteorological factors were

**132**  averaged from four measurements (scalar averaging for most factors and vector averaging for wind speed

**133**  and wind direction, which involved using the u (U) and v (V) components for averaging). The

**134**  meteorological station with a minimum distance from the city center was chosen.

**135**  **2.3 Lamb-Jenkinson circulation typing**

**136**  The Lamb-Jenkinson weather type approach (Lamb, 1972; Yarnal, 1993; Conway and Jones, 1998; Trigo

**137**  and Dacamara, 2000; Mckendry et al., 2006; Demuzere et al., 2009; Russo et al., 2014; Santurtún et al.,

**138**  2015; Pope et al., 2016; Liao et al., 2017) has been widely employed to classify synoptic circulation. On

**139**  the basis of the Lamb-Jenkinson method, the weather type circulation pattern for a given day is described

**140**  using the locations of the high- and low-pressure centers that identify the direction of the geostrophic

**141**  flow; the method uses coarsely-gridded pressure data on a 16-point moveable grid (Demuzere et al.,

**142**  2009). In our study, North China was set as the center. The specific schematic diagram is shown in Fig.

**143**  1a. The daily mean sea level pressure data were averaged over four time points to determine the daily

**144**  weather type. The detailed classification procedure can be found in Trigo and Dacamara (2000) and in

**145**  the supplementary information (Text S1).

**146**  **2.4 Reconstruction of O₃ concentration based on weather types**

**147**  To quantify the interannual variability captured by the variations in the surface circulation pattern,

**148**  Comrie and Yarnal (1992) suggested an algorithm to separate synoptic and non-synoptic variability in

**149**  environmental data; by multiplying the overall mean value of a particular pattern by the occurrence

**150**  frequency of that type of year, the climate signal can be obtained as follows:

**151**  $$\overline{\overline{O_{3\,m}}}(fre) = \sum_{k=1}^{26} \overline{O_{3k}} F_{km} \qquad (1)$$

**152**  where $\overline{\overline{O_{3\,m}}}(fre)$ is the reconstructed mean MDA8 O₃ concentration influenced by the frequency of

**153**  changes in the weather type from April-October for the year m, $\overline{O_{3k}}$ is the 5-year mean MDA8 O₃

**154**  concentration for weather type k, and $F_{km}$ is the occurrence frequency of weather type k during April-

**155**  October for year m.

**156**     Hegarty et al. (2007) suggested that variations in the circulation patterns are attributed to not only

**157**  frequency changes but also intensity variations; moreover, they noted that the environmental and climate-

**158**  related contributions to the interannual variations in ozone could be better separated by considering these

**159**  two changes. As a result, Equation (1) was modified into the following form:

**160**  $$\overline{\overline{O_{3\,m}}}(fre + int) = \sum_{k=1}^{26} (\overline{O_{3k}} + \Delta O_{3km}) F_{km} \qquad (2)$$

where $\overline{O_{3\,m}}(fre + int)$ is the reconstructed mean MDA8 $O_3$ concentration influenced by the frequency
and intensity of the changes in circulation patterns from April-October for year m; $\Delta O_{3km}$ is the
modified difference on the fitting line, which is obtained through a linear fitting of the annual MDA8 $O_3$
concentration anomalies ($\Delta O_3$) to the circulation intensity index (CII) for circulation pattern k in year m.
$\Delta O_{3km}$ represents the part of the annual observed ozone oscillation caused by the intensity in each
circulation pattern. Hegarty et al. (2007) used the domain-averaged sea level pressure (mslp) to represent
the CII.
To better characterize intensity variations, we used addition 5 CIIs: the difference between the highest
pressure and lowest pressure (gradient), the center pressure of the highest pressure system (max slp), the
center pressure of the lowest pressure system (min slp), the distance from the highest pressure centers to
the study city (dis max), and the distance from the lowest pressure centers to the study city (dis min).
Among the above 6 CIIs, that having the strongest correlation coefficient (r) with $\Delta O_3$ was selected as
an effective circulation intensity index (ECII). Thus, ECII was used in Equation (2) to calculate $\Delta O_{3km}$.
All CIIs for the 14 cities were calculated based on $10°×10°$ grids covering North China (32°N-42°N,
110°E-120°E). An example of $\Delta O_{3km}$ (weather type C in ZJK which is a city located in Hebei province)
is shown in Fig. 7a. Here, min slp has the highest r (-0.97) among the 6 CIIs in type C in ZJK, so min slp
is selected as the ECII.
**2.5 The segmented synoptic-regression approach and model validation**
The utilization of a segmented synoptic-regression approach can aid in minimizing the errors when
using linear regression to model a nonlinear relationship and effectively forecast ozone variations
(Robeson and Steyn, 1990; Liu et al., 2007; Demuzere and van Lipzig, 2010; Liu et al., 2012). Based on
locally monitored meteorological data, their 24-h time lag values and weather type classifications,
stepwise linear regression was used in every weather category to construct the ozone potential forecast
model. The details of the main methods are shown in Text S2. Notably, in this research, after excluding
the missing data and disordering the time sequences, 80% of these days were used to build the potential
forecast equations, and the remaining 20% were used to validate the accuracy of the equations.
Statistical model performances were evaluated according to the following factors: $R^2$ (variance in the
individual model's coefficients of determination), RMSE (root mean square error), and CV (coefficient
of variation defined as RMSE/mean MDA8 $O_3$). All statistics are based on MATLAB R2015b.
**3. Results and discussion**
**3.1 Characteristics and variation trend of ozone concentrations in North China**
The MDA8 $O_3$ concentration is one of six factors used to calculate the daily air-quality index in China.
Five ranks were separated, representing different air-quality levels: excellent, good, lightly polluted,
moderately polluted and heavily polluted days, with cut-off concentrations of 100, 160, 215 and 265 μg
m$^{-3}$, respectively. The daily limit for the Grade II National Ambient Air Quality Standard is 160 μg m$^{-3}$.
The spatial distribution of the averaged MDA8 $O_3$ concentration (Fig. 1b) and exceedance ratio, which
represents the proportion of days exceeding the standard (160 μg m$^{-3}$) (Fig. 1c), as well as detailed
information on the 58 cities (Table S1), showed a severe ozone pollution problem during the last five
years in North China. The domain-averaged MDA8 $O_3$ concentration for 58 cities was 122±11 μg m$^{-3}$,
with an increasing rate of 7.88 μg m$^{-3}$ year$^{-1}$ and an exceedance ratio of 22.2±8.2%. Notably, the most
polluted cities were concentrated in Beijing, the southeast of Hebei and the west and north of Shandong,
where the average MDA8 $O_3$ concentration was 130±9 μg m$^{-3}$ and the exceedance ratio was 27.9±7.2%.
The daily evolution of MDA8 $O_3$ concentrations in 14 cities from 2013-2017 (Fig. 2a) indicated
periodic, consistent and regional characteristics of ozone pollution. The most highly polluted periods
were from mid-May to mid-July. In particular, the frequency and level of ozone pollution increased
significantly in 2017, and the number of regionally persistent ozone pollution events increased. The rate
of the increase in the MDA8 $O_3$ concentration from 2013-2017 was 0.87 μg m$^{-3}$ month$^{-1}$ (Fig. 2b), and
this growth was accompanied by a decrease in the PM$_{2.5}$ concentration (Fig. S1). A reduction in particle
extinction due to a decreased PM$_{2.5}$ concentration can lead to an increase in radiation reaching the ground;
in addition, Li et al. (2019) suggested that decreased PM$_{2.5}$ concentrations slowed the sinking of
hydroperoxyl (HO$_2$) radicals and thus stimulated ozone production. Thus, the rise in ozone was partly
due to the decline in PM$_{2.5}$. Overall, the annual domain-averaged MDA8 $O_3$ concentrations for 58 cities
were 102, 109, 116, 119 and 136 μg m$^{-3}$ in 2013, 2014, 2015, 2016 and 2017, respectively (Fig. 3a). The
exceedance ratios for all cities were found to be 12.9%-19.4% from 2013 to 2016 but reached 31.1% in
215 2017.
The monthly-mean MDA8 $O_3$ concentrations (Fig. 3b) from April to October were 112, 138, 149, 132,
124, 117 and 75 μg m$^{-3}$, respectively, and the corresponding exceedance ratios were 9.4, 30.1, 41.1, 26.1,
20.3 20.1 and 3.3%. The highest domain-averaged MDA8 $O_3$ concentration and exceedance ratio
occurred in June, followed by those in May, July, August, September, April and October. Meteorological
conditions led to high ozone concentrations in June, and monsoon circulation in July and August resulted
in cloudy, rainy conditions and less radiation in the study area (Wang et al., 2009c; Tang et al., 2012).
The higher ozone concentrations in April than compared with those in October could be associated with
strong winds, resulting in a downward transport of ozone due to the lower stratosphere folding
mechanism (Stohl and Trickl, 1999; Cooper et al., 2002; Delcloo, 2008; Verstraeten et al., 2015). Notably,
this conclusion is different from that of Tang et al. (2012), who reported that the ozone concentration in
July was higher than that in May in North China during 2009-2010. However, as our study indicated that
the domain-averaged MDA8 $O_3$ in May was even higher than that in July, the concentrated pollution
episode occurred earlier, especially in 2017. The second half of May was the most polluted period, when
the exceedance ratio was 46.1%, which is higher than the ratios observed in the first half of June (39.5%),
the second half of June (45.4%) and the first half of July (35.6%). The reason for this difference is
probably the abnormally high temperatures in May, especially the second half of May, from 2013-2017
and particularly in 2017 (Fig. S2). Many studies have found a strong positive correlation between ozone
levels and temperature (Bloomer et al., 2009; Demuzere et al., 2009; Bloomer et al., 2010; Pusede et al.,
234 2015).

**3.2 Weather types and associated surface $O_3$ levels**

**3.2.1 The meteorological conditions and regional ozone concentrations under different predominant weather types**

Based on the Lamb-Jenkinson weather typing technique, 26 circulation patterns affecting North China
were identified, including two vorticity types (anticyclone, A, and cyclone, C), eight directional types
(northeasterly, NE; easterly, E; southeasterly, SE; southerly, S; southwesterly, SW; westerly, W;
northwesterly, NW; and northerly, N) and 16 hybrids of vorticity and directional types (CN, CNE, CE,
CSE, CS, CSW, CW, CNW, AN, ANE, AE, ASE, AS, ASW, AW, and ANW). The composite mean sea

level pressure maps, along with the occurrence days, are shown in Fig. 4. There are distinctly different locations of the high-pressure and low-pressure centers under the different circulation conditions. The occurrence ratios of vorticity types, pure directional types, and hybrid types were 35.6%, 38.8% and 25.6%, respectively, during all 1070 days.

The mid-latitude eastern Eurasian continent is strongly affected by monsoon circulation, and there are several key synoptic systems affecting the circulation and meteorological conditions in North China. During our study period, North cyclones (Mongolian and Yellow River cyclones), which are indicative of a low-pressure system located in northwest North China, dominated in spring and summer. The Siberian High influenced northern China in spring and autumn. The Western Pacific Subtropical High was also a key system in summer. Therefore, these main synoptic systems resulted in variations in the frequencies of the various weather types in different months over North China.

According to the different locations of the different central systems, together with the similar meteorological factors and mean MDA8 $O_3$ values in these circulation patterns, 26 circulation types were merged into 5 weather categories: 1) N-E-S direction (geostrophic wind direction diverts from north to south) including N, NE, E, SE, AN, ANE, AE and ASE; 2) S-W-N direction (geostrophic wind direction diverts from south to north)including S, SW, W, NW, AS, ASW, AW and ANE; 3) LP (low-pressure related weather types) including CN, CNE, CE, CSE, CS, CSW, CW and CNW; 4) A (anticyclone); and 5) C (cyclone). The occurrence ratios of the 5 weather categories were 25.4%, 26.5%, 12.5%, 17.5% and 18.1% in all 1070 days. The predominant local meteorological conditions associated with a specific weather category play an important role in ozone pollution, influencing ozone photoreaction or its regional transport. The values of the averaged MDA8 $O_3$ concentration, frequency of weather types/categories and meteorological variables are depicted in Table 1 and Fig. 5. Briefly, the N-E-S direction and A categories were typically associated with cool and wet air, moderate rain and TCC, low BLH, and relatively clean air masses from the Inner Mongolia/eastern ocean region (Fig. S3); these conditions are unfavorable for ozone formation; thus, the corresponding area-averaged MDA8 $O_3$ concentrations were 98±6 μg m$^{-3}$ and 96 μg m$^{-3}$, respectively. The S-W-N direction category had moderate T and BLH, low RH, weak wind, sporadic clouds and rain, and strong subsidence in the lower troposphere, which contributed to high ozone levels (122±8 μg m$^{-3}$). The highest ozone concentrations (126±16 and 128 μg m$^{-3}$) were related to the LP and C categories, which can probably be attributed to the meteorological conditions (hot and humid air, a small amount of TCC and rainfall, and high BLH) that were favorable for ozone formation and transport. However, CE and CSE were different from the other weather types in the LP category, with low $O_3$ concentrations due to low temperatures and easterly winds from the ocean. Overall, the peak values of ozone always occurred in the front of the passage of a cold front or cyclone (most weather types in LP and C), whereas the lowest values occurred during or after the passage of a cold front (most weather types in the N-E-S direction, C with heavy rainfall and CE); similar conclusions were also previously reported (Cooper et al., 2001; Cooper et al., 2002; Chen et al., 2008)

**3.2.2 Spatial distributions of the 26 weather types/five categories**

The spatial distribution of the averaged MDA8 $O_3$ concentration under different weather types is shown in Fig. 6, and Figs. S3-S7 display the spatial distributions of the combined wind field with BLH, maximum temperature (Tmax), RH, rain and TCC, respectively. In most cities, the lowest MDA8 $O_3$ concentrations occurred in the N-E-S direction and A weather categories. The S-W-N direction category, having predominantly southerly winds throughout the region or south of North China, exhibited high

ozone values along with the prevailing wind direction. The LP and C weather categories, having the highest regional averaged levels, were associated with high Tmax and strong southerly flow, moderate RH and ample sunshine, which are the meteorological conditions that are favorable for ozone formation as well as the transport of ozone and its precursors from polluted areas.

### 3.2.3 Interannual/monthly ozone variation elaborated from the perspective of circulation pattern changes

The interannual or monthly ozone concentration changes are associated with variations in weather types. Fig. 3 indicates that the ratios of high-ozone weather categories (S-W-N direction, LP and C S-W-N direction, LP and C) were most frequent in 2013 and 2017, less frequent in 2015 and 2016, and least frequent in 2014. The high-ozone weather categories accounted for 61.5% and 61.8% of the time in 2013 and 2017, respectively. Under similar weather conditions, low ozone levels could also be associated with high $PM_{2.5}$ levels in 2013 and 2015. The contributions of frequency-only and circulation changes (frequency and intensity) to the interannual ozone variability will be discussed in Section 3.3.

Due to the impacts by monsoon circulation systems, the frequencies of weather types varied dramatically on a monthly scale (Fig. 3b). The frequencies of both the N-E-S direction and A gradually decreased in spring, whereas the frequencies of the S-W-N direction, LP and C gradually increased. In autumn, the frequencies of LP and C decreased, whereas those of the S-W-N direction, N-E-S direction and A increased. The weather categories C and LP dominated in summer. The high-ozone weather categories (S-W-N direction, LP and C) accounted for 58.7, 66.5, 79.3, 80.6, 49.0, 38.0 and 27.7% of the time in the months from April to October, respectively. These frequencies were highest in July, June, and May, which probably resulted in the highest monthly averaged regional ozone concentrations. However, due to the influence of monsoon circulation, large amounts of rainfall occurred during July: 73 out of 194 days during the 5 years were rainy in category C, which reduced the ozone levels. Notably, severe ozone pollution in May, especially in the second half of May in 2017, was closely related to abnormally high frequencies under the control of the most polluted synoptic categories (LP and C), accounting for 35.5% in 31 days and 50.0% in 16 days (Table S2). With the development of the Siberian High from August to October, the N-E-S direction and A weather categories occurred frequently, and the monthly averaged ozone concentrations declined.

### 3.3 Effects of synoptic changes on interannual ozone variability

### 3.3.1 Effect of weather type intensity on interannual ozone variability

The pressure fields for the 26 synoptic types per year from 2013-2017 (Figs. S8-S9) indicated that every synoptic weather type varied in both frequency and intensity. The intensity of the circulation patterns indicated the differences in the center pressure, the location of the predominant system, the pressure gradient, and the domain-averaged sea level pressure. The correlations between ECII and $\Delta O_3$ (as introduced in Section 2.4) differed in the different circulation types in the various cities. For instance, the strong negative correlation between these two variables for weather type C in ZJK (Fig. 7a) indicated that the low values of min slp were associated with high MDA8 $O_3$ concentrations.

The number of cities and averaged r values according to the corresponding ECII under each circulation type among the 14 cities are shown in Fig. 7b. Overall, the average absolute value of r was 0.74. For circulation type C, $\Delta O_3$ was strongly correlated with min slp in 9 of the cities, and the average r was -0.81, i.e., a strong negative correlation. A strong negative correlation between $\Delta O_3$ and the pressure

gradient was evident for circulation type N, whereas an opposite pattern occurred for circulation types CSE and CS. The reasons for this difference are as follows. Northerly winds prevailed for circulation type N, and high-pressure gradients indicated strong northerly winds that brought clean air masses from the north, which resulted in a decrease in the MDA8 $O_3$ concentration. However, high temperatures and RH as well as prevailing southerly or easterly winds (Figs. S3-S5) occurred in southern cities in the CSE and CS circulation types. In addition, the abundance of precursors and ozone in the upwind region facilitated ozone formation and transport with the increasing pressure gradient (wind speeds).

Even under the same weather type controls, the ECII and the values of r differed in the different cities. This phenomenon was caused by differences in geographic location, topographic discrepancies, and the properties of the upwind air mass. Therefore, under the control of the same weather type, the ECII was the same in adjacent cities.

### 3.3.2 Quantifying the effects of the interannual synoptic changes on the interannual ozone variability

Based on Equations (1) and (2), we reconstructed the interannual ozone levels by taking into account either frequency-only or both frequency and intensity variations in synoptic circulations, which are $\overline{\overline{O_{3m}}}(\text{fre})$ and $\overline{\overline{O_{3m}}}(\text{fre}+\text{int})$, respectively. The differences between the maximum and minimum annual reconstructed ozone are labeled as $\Delta\overline{\overline{O_{3m}}}(\text{fre})$ and $\Delta\overline{\overline{O_{3m}}}(\text{fre}+\text{int})$, respectively. $\Delta O_3\_\text{obs}$ differed between the maximum and minimum for the annual observed $O_3$ concentration. Thus, the contributions of interannual variability in $O_3$ influenced by frequency-only and frequency and intensity variations in synoptic circulation were $\Delta\overline{\overline{O_{3m}}}(\text{fre})/\Delta O_3\_\text{obs}$ and $\Delta\overline{\overline{O_{3m}}}(\text{fre}+\text{int})/\Delta O_3\_\text{obs}$, respectively, which indicate the interannual oscillations in ozone levels caused by synoptic variability.

The observed and reconstructed (influenced by frequency-only and frequency and intensity variations in synoptic circulations) interannual MDA8 $O_3$ levels for 5 years in 14 cities and the whole region are shown in Fig. 8. The contributions of interannual variability in $O_3$ influenced by frequency and intensity variations in synoptic circulation ranged from 44.1 to 69.8% over the 14 cities, and the contributions by frequency-only variations ranged from 5.2 to 23.4%. Obviously, the interannual fluctuations in the ozone concentration were caused mainly by weather type intensity changes in North China. In addition, based on the regional averaged scale, the interannual variability in the domain-averaged observed MDA8 $O_3$ in 14 cities varied from averaged maximum values of 135 μg m$^{-3}$ in 2017 to a minimum of 104 μg m$^{-3}$ in 2013. The contributions of variations in circulation patterns to interannual $O_3$ increases were 39.2%, and the remaining interannual variability was possibly due to nonlinear relationships resulting from recent emission control measures over North China.

In most cities, the contributions of synoptic circulation changes on ozone variability obtained here (44.1-69.8%) are higher than that (50%) estimated by Zhang et al. (2013). The difference could be attributed to our results being based on (1) more weather types, (2) weather types covering all days, and (3) more CIIs, which can better characterize the intensity of slp. Furthermore, a higher contribution in single city and increasing reconstructed ozone indicate that synoptic circulation patterns play an important role in the ozone variability in North China. However, our regional contribution (39.2%) is lower than that (46%) estimated by Hegarty et al. (2007), which reveals that the increasing trend of ozone concentrations from 2013 to 2017 in North China is largely associated with the impact of its precursors.

### 3.4 Quantifying the impact of weather patterns on day-to-day ozone concentration and forecasting daily ozone concentration

Based on the five weather categories defined in Section 3.2.1, a segmented synoptic-regression analysis approach (introduced in Section 2.5) was established to quantify the impact of weather patterns on the day-to-day ozone concentration and to construct the ozone potential forecast model.

The contributions of local meteorological factors to the day-to-day variations in ozone can be evaluated by the explained variance ($R_E^2$) calculated from the synoptic-regression-based models (Hien et al., 2002; Wang et al., 2009b). Overall, the predicted versus observed MDA8 O$_3$ concentrations for the validation data are shown in Fig. 9; the predicted concentrations were obtained by inputting the validation data (the part that was not used to build the model, which was 20% of the total data) into the corresponding model equations for the five weather categories for each city. Local meteorological parameters explained 57-63% and 41-52% of the day-to-day variability in the MDA8 O$_3$ concentration for the northern cities (except for QHD, 34%) and southern cities (except for ZZ, 20%), respectively.

In addition, the results of segmented synoptic-regression analysis in 14 cities, i.e., the daily MDA8 O$_3$ potential forecast equations for each category in each city, are shown in Table S3. Table 2 represents the number of cities (14 total) from which the meteorological factors were used in a stepwise regression model under each weather category. The results show that Tmax exhibited a strong positive correlation with ozone; thus, this factor is the primary influencing factor in all categories and all cities, as high temperatures are related to high ozone concentrations in North China. V exhibited a positive correlation with O$_3$ in the northern part of this region (at the north-south boundary at approximately 38.5°N), which means that southerly flow caused an increase in the ozone concentration. Therefore, as discussed in Section 3.2.2, high temperatures and southerly winds were the main factors that contributed to increased ozone concentrations in North China from a regional perspective. Both RH_lag and RH showed a negative correlation with O$_3$, and the former had more occurrences and greater weights in the equations than the latter. This phenomenon may exist because RH of approximately 40-50% (Zhao et al., 2019) generates more hydroxyl radicals (OH), facilitating ozone formation, and ozone is stored in the residual layer and transported to the surface the next day via convection and diffusion. In addition, TCC is a key factor.

Three statistical measures ($R^2$, RMSE, and CV) for the building and validation datasets for the 5 weather categories and the composite model, which integrates the 5 weather categories, in the 14 cities (Table S4) indicates that the potential forecast equations for MDA8 O$_3$ were acceptable in most cities in most cities. Scatterplots of predicted versus observed MDA8 O$_3$ concentrations in the composite validation datasets in each city are shown in Fig. 9. For the validation data, the prediction of ozone concentration was obtained by inputting the meteorological factors into the simulated formula for the corresponding weather category in each city; therefore, the composite validation datasets indicated the integrated predicted ozone concentrations for the five categories. The results of validation show that $R^2$ was higher than 0.50, except for QHD, SJZ and ZZ (0.24-0.47), while CV was lower than 40%, except for TY and ZZ.

The results reveal that most of the validation data are within the acceptable error range, as they are concentrated within the 2:1 and 1:2 ratio lines, and the scatters are distributed evenly around the 1:1 line. For example, the comparison of the observed and predicted ozone in Beijing during our study period is shown in Fig. S10. This finding also indicates that the segmented synoptic-regression approach is practical for constructing ozone potential forecasting models in most cities in North China.

In brief, the aforementioned results can provide references for daily MDA8 O$_3$ prediction for each city
and facilitate the understanding and evaluations of the impact of local meteorology on daily ozone
variations on an urban scale.
**4 Conclusions**
In this study, we demonstrated the interannual/monthly variations in the surface MDA8 O$_3$ concentration
in North China during April-October 2013-2017, investigated the relationship between weather types
and MDA8 O$_3$ levels, quantified the contributions of weather types and local variations in meteorological
factors to both the interannual and day-to-day variability in ozone, and built ozone potential forecast
equations. The main results are as follows:
1. The annual domain-averaged concentrations of MDA8 O$_3$ during 2013-2017 were 102, 109, 116,
119, and 136 μg m$^{-3}$, respectively, and the highest exceedance ratio (31.1%) was observed in 2017. The
monthly mean MDA8 O$_3$ concentrations were 112, 138, 149, 132, 124, 117, and 75 μg m$^{-3}$ from April to
October, respectively, with a significantly increasing rate of 0.87 μg m$^{-3}$ month$^{-1}$ during the five-year
period. The most polluted cities were concentrated around Beijing, the southeast of Hebei and the west
and north of Shandong.
2. Twenty-six weather types were objectively identified based on the Lamb-Jenkinson method and
combined into five weather categories according to similar meteorological factors and MDA8 O$_3$
concentrations. The high ozone levels in 2017 and during May-July were partly due to the high frequency
of the highly polluted weather categories (S-W-N direction, LP and C) resulting from high temperatures,
moderate RH and southerly air flows.
3. The intensity of synoptic circulation patterns was the dominant factor through which variations in
weather types influenced the variability in the ozone levels. The contributions of interannual variability
in O$_3$ influenced by both frequency and intensity variations in synoptic circulation patterns ranged from
44 to 70% over the 14 cities that were evaluated in detail, whereas the contributions of the variations in
circulation patterns to the increase in the interannual O$_3$ from 2013 to 2017 was only 39.2% based on a
regionally averaged scale.
4. The results of the daily ozone potential forecast equations in the 14 cities showed that high
temperatures, moderate RH and southerly winds could result in severe ozone pollution in the northern
part of North China, whereas the southern part was mainly affected by high temperatures and RH. Local
meteorological parameters explained 55-64% and 43-49% of the day-to-day MDA8 O$_3$ variability for
the northern cities (except for QHD, 32%) and southern cities (except for ZZ, 25%), respectively.
**Author contribution**
LL Wang designed this research. JD Liu and LL Wang interpreted the data and wrote the paper. MG Li
processed some of the data. The weather type classification program was provided by ZH Liao. Y Sun,
T Song, and WK Gao provided some of the PM$_{2.5}$ and O$_3$ data. Y Li provided some of the meteorological
data. All of the authors commented on the paper.

**Data availability**

The daily average mass concentrations of ozone were obtained from the National Urban Air Quality Real-time Publishing Platform (http://106.37.208.233:20035/) issued by the Chinese Ministry of Ecology and Environment. Daily meteorological data were obtained from the China Meteorological Administration in the Meteorological Information Combine Analysis and Process System (MICAPS), and daily meteorological reanalysis data (gridded at $1° \times 1°$) were obtained from ERA-Interim (https://apps.ecmwf.int/datasets/data/interim-full-daily/levtype=sfc/). All of the data can be obtained upon request.

**Competing interests**

The authors declare that they have no conflict of interest.

**Acknowledgments**

This work was partially supported by grants from the National Key R&D Plan (Quantitative Relationship and Regulation Principle between Regional Oxidation Capacity of Atmospheric and Air Quality 2017YFC0210003), the National Natural Science Foundation of China (No. 41505133 & 41775162), the Strategic Priority Research Program of the Chinese Academy of Sciences (No. XDA19020303), the National Research Program for Key Issues in Air Pollution Control (DQGG0101), the Postgraduate Research & Practice Innovation Program of Jiangsu Province (No. 1344051901061) and a program of the China Scholarships Council. We give special thanks to the National Earth System Science, Data Sharing Infrastructure, and the National Science & Technology Infrastructure of China.

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

**Tables**
**Table 1. Weather types, ozone concentrations and meteorological conditions for 5 weather categories.**

| category | type | ozone | fre | Tmax | RH | rain | TCC | ws | BLH | div | v-v | characteristics |
|---|---|---|---|---|---|---|---|---|---|---|---|---|
| N-E-S direction | N | 108 | 5.4 | 25.9 | 64.5 | 2.2 | 6 | 2.1 | 749 | 0.85 | 2.28 | MDA8 $O_3$ (98±6 µg m$^{-3}$). Cool, moderate rain, moderate TCC, and low BLH, predominant wind directions are north and east, clean air masses from inner Mongolia or the eastern ocean. |
| | NE | 94 | 6.4 | 25.5 | 72.1 | 5.0 | 7 | 2.2 | 637 | -1.01 | -0.70 | |
| | E | 98 | 4.7 | 25.4 | 70.5 | 3.4 | 6 | 2.1 | 618 | -1.34 | -1.85 | |
| | SE | 105 | 2.3 | 22.8 | 71.5 | 4.9 | 7 | 2.4 | 612 | -1.25 | -3.85 | |
| | AN | 101 | 1.7 | 22.2 | 61.2 | 1.5 | 5 | 2.3 | 738 | 2.36 | 5.12 | |
| | ANE | 88 | 2.4 | 23.1 | 67.9 | 2.3 | 6 | 2.2 | 681 | 0.79 | 1.53 | |
| | AE | 94 | 1.3 | 23.2 | 65.8 | 2.2 | 7 | 2.3 | 618 | -0.10 | 0.12 | |
| | ASE | 99 | 1.1 | 22.4 | 71.3 | 2.3 | 7 | 2.2 | 578 | 1.10 | 0.03 | |
| S-W-N direction | S | 112 | 4.1 | 25.4 | 65.7 | 2.2 | 6 | 2.2 | 642 | 0.28 | -1.23 | MDA8 $O_3$ (122±8 µg m$^{-3}$). Moderate T and BLH, lower RH, weak wind, sporadic clouds and precipitation, divergence in low troposphere. Prevailing southerly and westerly winds. |
| | SW | 131 | 6.2 | 26.5 | 60.3 | 0.6 | 5 | 2.1 | 716 | 1.81 | 1.34 | |
| | W | 133 | 5.4 | 26.6 | 58.3 | 1.0 | 5 | 2.2 | 763 | 2.33 | 2.43 | |
| | NW | 124 | 4.2 | 26.8 | 58.7 | 1.6 | 6 | 2.3 | 835 | 1.66 | 3.45 | |
| | AS | 120 | 1.4 | 24.8 | 63.3 | 0.7 | 6 | 2.0 | 641 | 1.76 | 0.54 | |
| | AS | 114 | 2.6 | 24.5 | 62.2 | 0.7 | 6 | 1.9 | 666 | 2.53 | 1.02 | |
| | AW | 126 | 1.0 | 23.8 | 58.5 | 0.2 | 5 | 1.8 | 685 | 3.14 | 4.16 | |
| | AN | 115 | 1.5 | 23.4 | 55.2 | 0.9 | 6 | 2.3 | 794 | 2.47 | 5.07 | |
| LP | CN | 135 | 2.0 | 29.8 | 68.0 | 2.2 | 6 | 1.9 | 732 | 0.09 | 0.92 | The hybrid of cyclone and direction types, MDA8 $O_3$ (126±16 µg m$^{-3}$). Widespread hot, humid, a small amount of clouds and rain, comparatively high BLH. |
| | CNE | 119 | 1.8 | 28.2 | 66.0 | 3.2 | 6 | 2.2 | 724 | -1.15 | -0.19 | |
| | CE | 109 | 0.8 | 25.4 | 73.6 | 6.4 | 7 | 2.1 | 559 | -2.67 | -3.65 | |
| | CSE | 103 | 0.7 | 25.1 | 62.6 | 1.4 | 6 | 2.6 | 725 | -1.65 | -0.58 | |
| | CS | 123 | 1.0 | 27.4 | 65.7 | 1.6 | 5 | 2.1 | 693 | -0.40 | -0.67 | |
| | CS | 155 | 2.2 | 29.4 | 62.6 | 1.2 | 5 | 2.3 | 796 | 0.96 | 0.58 | |
| | CW | 140 | 2.6 | 28.6 | 62.3 | 1.3 | 5 | 2.2 | 778 | 0.95 | 0.93 | |
| | CN | 124 | 1.5 | 29.2 | 62.4 | 4.5 | 6 | 2.5 | 853 | 0.12 | 1.26 | |
| C | | 128 | 18.1 | 29.5 | 67.1 | 3.8 | 6 | 2.2 | 715 | -1.20 | -1.22 | Cyclone, similar to LP. |
| A | | 96 | 17.5 | 22.3 | 64.5 | 1.5 | 6 | 2.0 | 632 | 2.54 | 2.66 | Anticyclone, similar to N-E-S direction. |

**Note: Ozone, MDA8 $O_3$ concentration (µg m$^{-3}$); fre, frequency of each type (%); Tmax, daily maximum**
**temperature (°C); RH, relative humidity (%); rain, total daily precipitation (mm); TCC, total cloud cover;**
**WS, wind speed (m s$^{-1}$); BLH, boundary layer height (m); div, divergence of the wind field (10$^{-6}$ s$^{-1}$) from 1000**
**to 850 hPa (7 levels); and v-v, vertical velocity from 1000 to 100 hPa (10$^{-2}$ Pa s$^{-1}$).**
**Table 2 All parameters used in the stepwise regression and the number of cities (out of 14) for which each**
**variable was selected for each weather category.**

| factors | N-E-S direction | S-W-N direction | LP | C | A |
|---|---|---|---|---|---|
| RH (%) | 2 | 4 | 3 | 4 | 6 |
| Tmax (°C) | 14 | 14 | 14 | 14 | 14 |
| rain (mm) | 0 | 2 | 2 | 0 | 2 |
| U | 2 | 3 | 3 | 3 | 1 |
| V | 9 | 6 | 6 | 7 | 2 |
| wd (°) | 2 | 0 | 0 | 1 | 1 |
| ws (m s$^{-1}$) | 4 | 1 | 1 | 0 | 2 |
| pre (hPa) | 0 | 0 | 0 | 0 | 0 |
| RH_lag (%) | 11 | 5 | 5 | 4 | 4 |
| Tmax_lag (°C) | 3 | 0 | 0 | 1 | 0 |
| rain_lag(mm) | 1 | 0 | 0 | 0 | 0 |
| U_lag | 2 | 1 | 1 | 1 | 1 |
| V_lag | 4 | 0 | 0 | 0 | 4 |
| wd_lag (°) | 0 | 0 | 0 | 0 | 0 |
| ws_lag (m s$^{-1}$) | 1 | 3 | 3 | 1 | 4 |
| pre_lag (hPa) | 1 | 0 | 0 | 1 | 1 |

**Note: RH, Tmax, rain, U, V, wd, ws and pre are relative humidity, maximum temperature, precipitation, u**
**component, v component, wind direction, wind speed and pressure, respectively. The suffix 'lag' means the**
**meteorological factors from the previous day.**

**Figures**

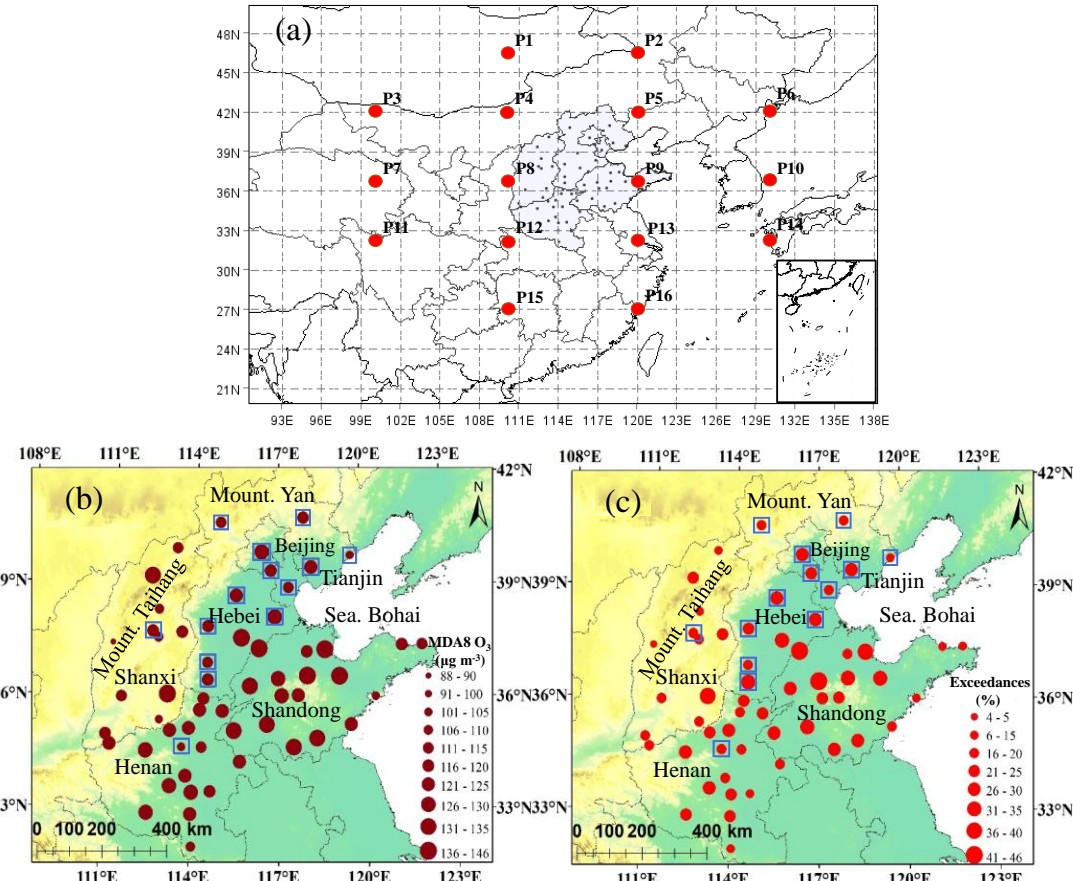


**Fig. 1. Location of North China (shaded area), all cities (black dots) and sea level pressure grids (a). The 16**
**red points show the locations of the 5°×10° mean sea level pressure grids used for the Lamb-Jenkinson**
**weather type classification. The spatial distributions of the maximum daily 8-h running average O₃ (MDA8**
**O₃) concentration (b) and exceedance ratios (c) for 58 cities. Statistics for 2013-2017 are shown in blue boxes;**
**the other boxes are those for 2015-2017. The base map is topography; the elevations of the Taihang Mountains**
**are more than 1200 meters, and the Yan Mountains range from 600 to 1500 meters.**

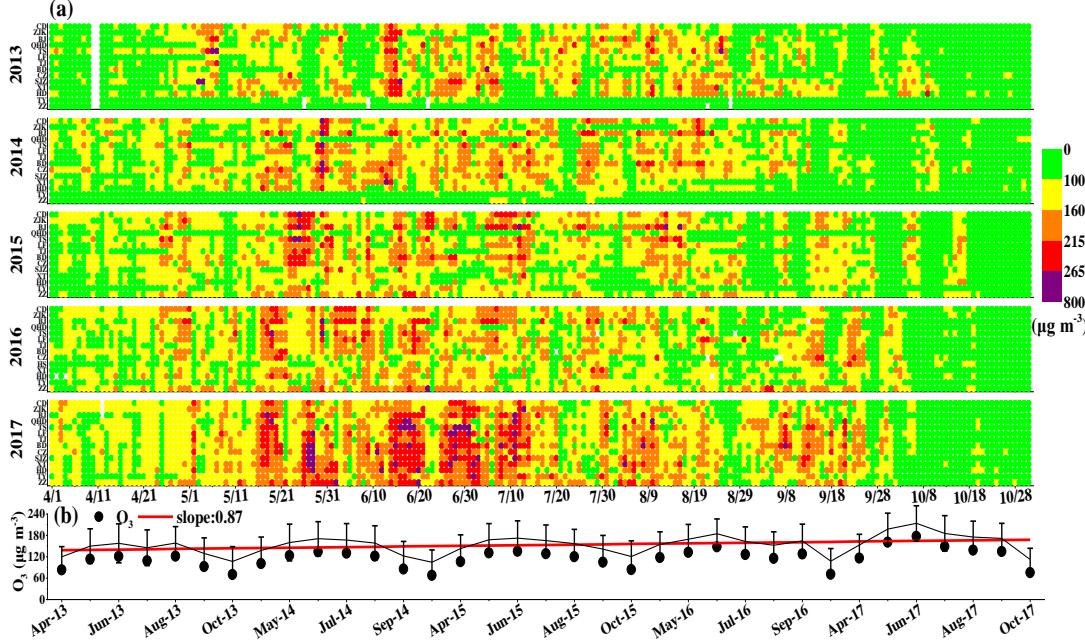


**Fig. 2.** Time series of daily MDA8 O₃ concentrations in 14 cities (north to south) (a), together with monthly averaged concentrations and standard deviations (b), during April to October from 2013 to 2017. Five ranks represent different air-quality levels, including excellent (green spots), good (yellow), lightly polluted (orange), moderately polluted (red) and heavily polluted (purple) days with cut-off concentrations of 100, 160, 215, and 265 µg m⁻³, respectively. The fit line (red line) in (b) represents the increasing trend of monthly mean MDA8 O₃.

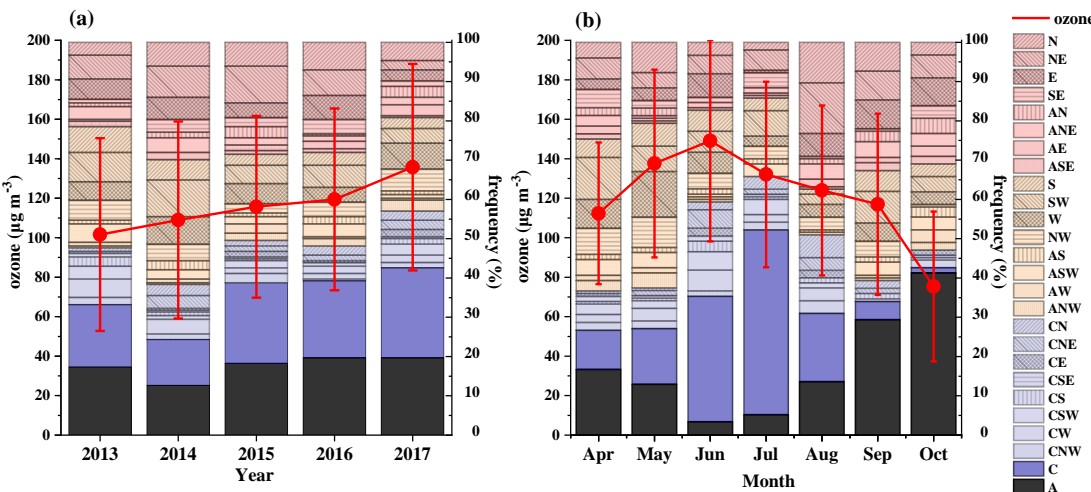

645

**Fig. 3.** Interannual (a) and monthly (b) averaged concentrations of ozone and frequencies of 26 **weather types** from April-October 2013-2017. The red dots represent the mean values, the vertical red lines indicate the standard deviations, and stacked charts represent the percentages of various **weather types** (2013 and 2014 are averaged for 14 cities; 2015-2017 are averaged for 58 cities). The pink, orange, light blue, dark blue and black areas represent the weather categories N-E-S direction, S-W-N direction, LP, C and A, respectively.

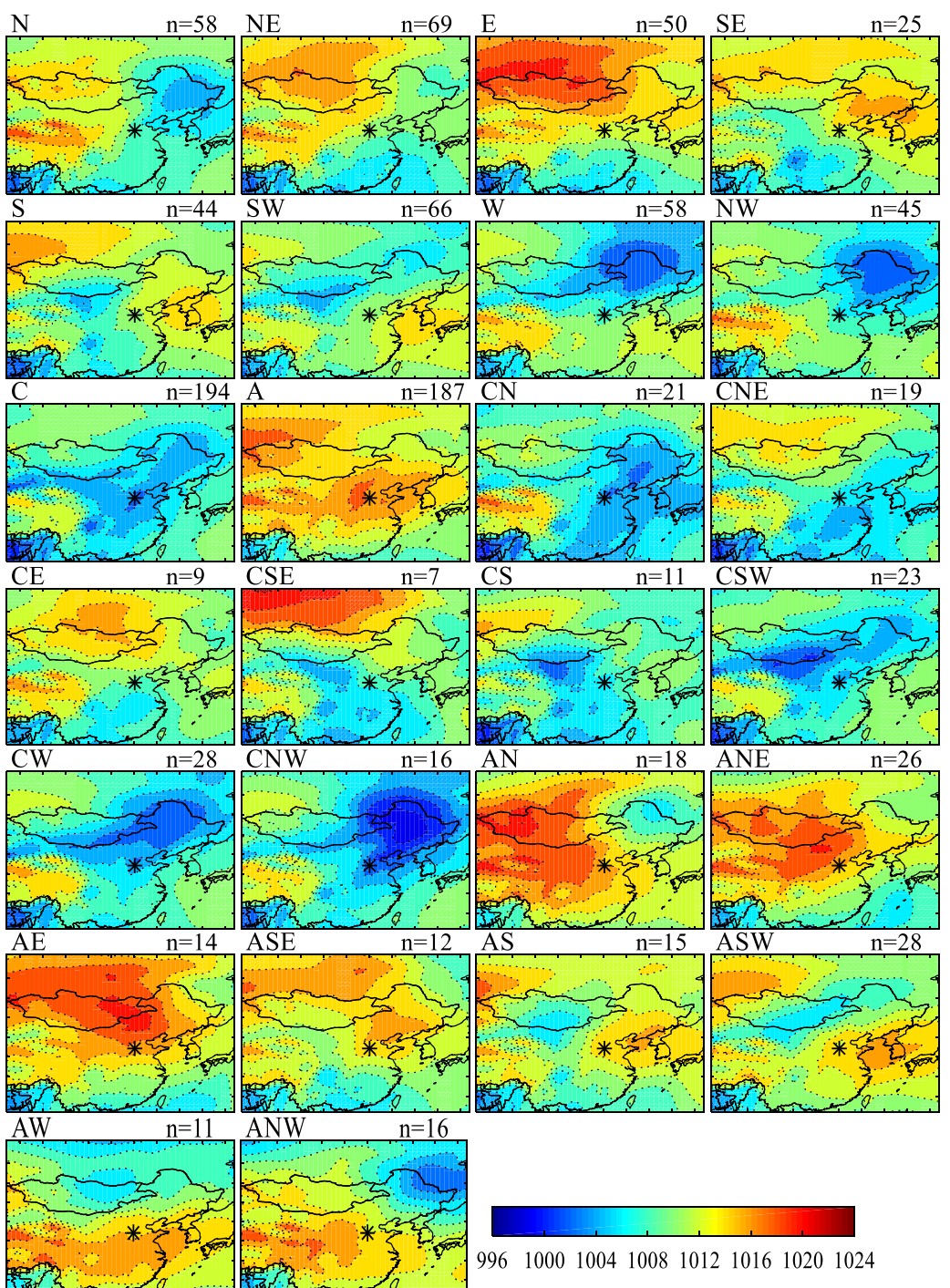


**Fig. 4. Mean surface pressure field (unit: hPa) for the 26 weather types during April-October of 2013-2017**
**and occurrence days (1070 days in total). '*' indicates the center of North China.**

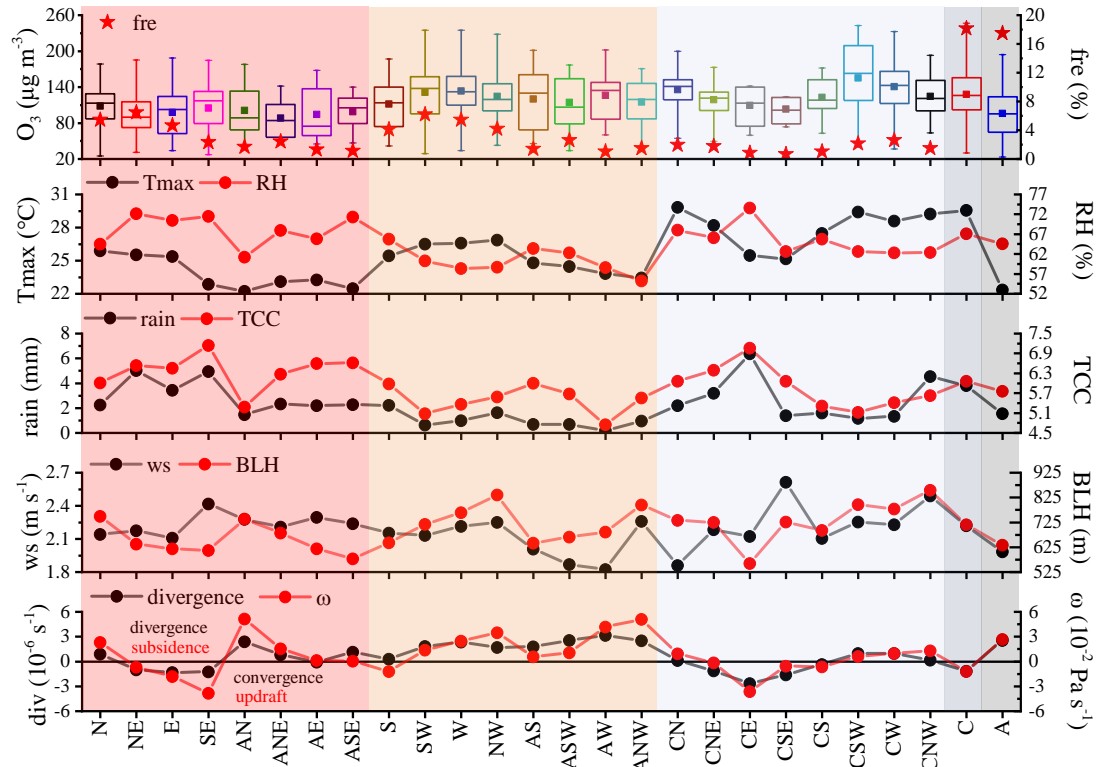


Fig. 5. Box chart of domain-averaged MDA8 O₃ concentrations, occurrence frequency of weather types (fre), and mean values of meteorological factors in 26 weather types during April-October 2013-2017. In the box chart, the solid square indicates the mean, the horizontal lines across the box are the averages of the first, median, and third quartiles, respectively, and the lower and upper whiskers represent the 5th and 95th percentiles, respectively. The pink, orange, light blue, dark blue and black areas represent the weather categories N-E-S direction, S-W-N direction, LP (low-pressure related weather patterns), C (cyclone) and A (anticyclone), respectively.

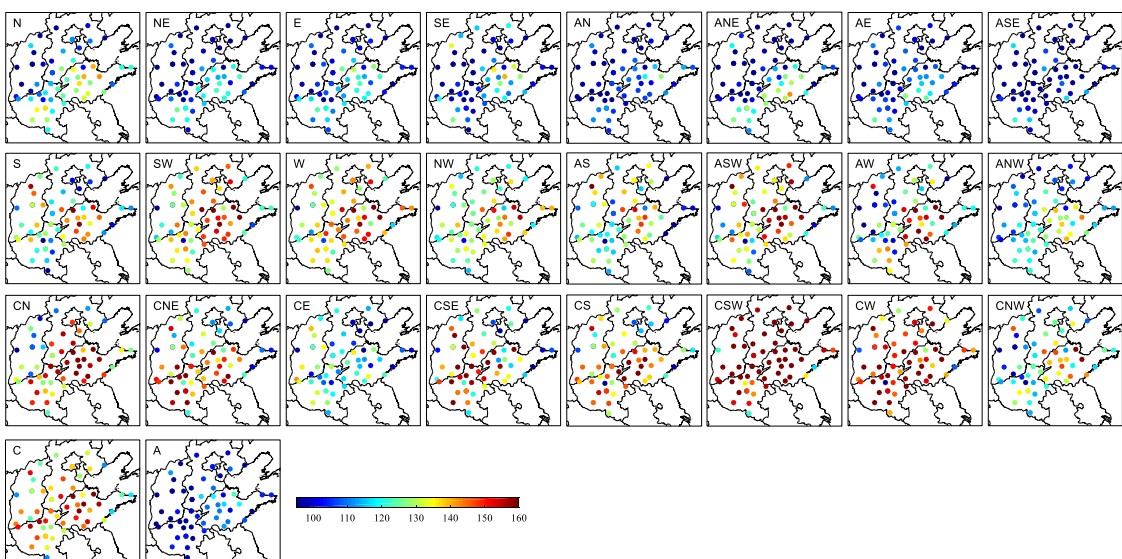

662

Fig. 6. Spatial distribution of average MDA8 O₃ for the 26 weather types. The first, second, and third rows correspond to the weather categories N-E-S direction, S-W-N direction and LP, respectively, and the fourth row includes both categories C and A.

666

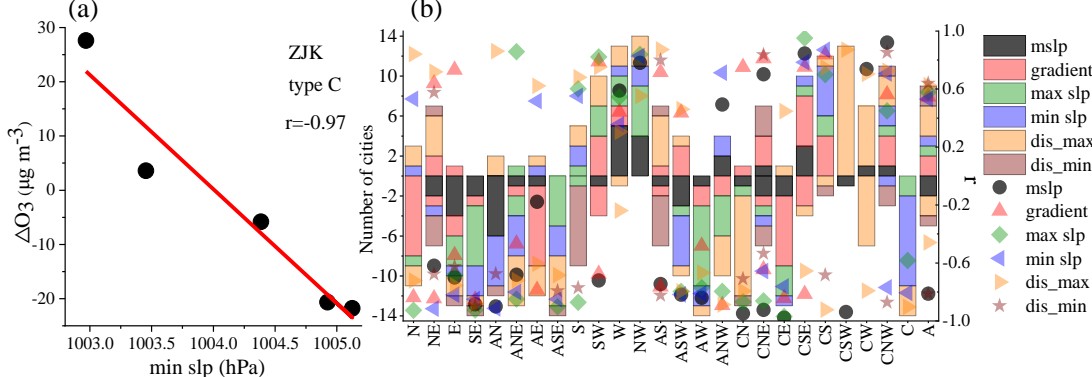

667

Fig. 7. Scatterplot of $\Delta O_3$ versus min slp for **weather type** C in ZJK (a). The red line represents the linear fitting between min slp (the ECII under **weather type** C in ZJK) and $\Delta O_3$ (the difference between the MDA8 $O_3$ for a given year and the corresponding 5-year average); r represents **the** correlation coefficient. The number of cities (histogram) and averaged correlation coefficient r (points of different shapes) according to corresponding ECII under each **weather type** among 14 cities (b). The number of cities with positive/negative values represents positive/negative correlations between ECII and $\Delta O_3$. For example, under CW controls, there are 1, 6 and 7 cities where ECII corresponds to mslp with **a** positive correlation, dis max with **a** positive correlation, and dis max with **a** negative correlation, respectively, and the average r is 0.74 and 0.70 and -0.79, respectively.

677

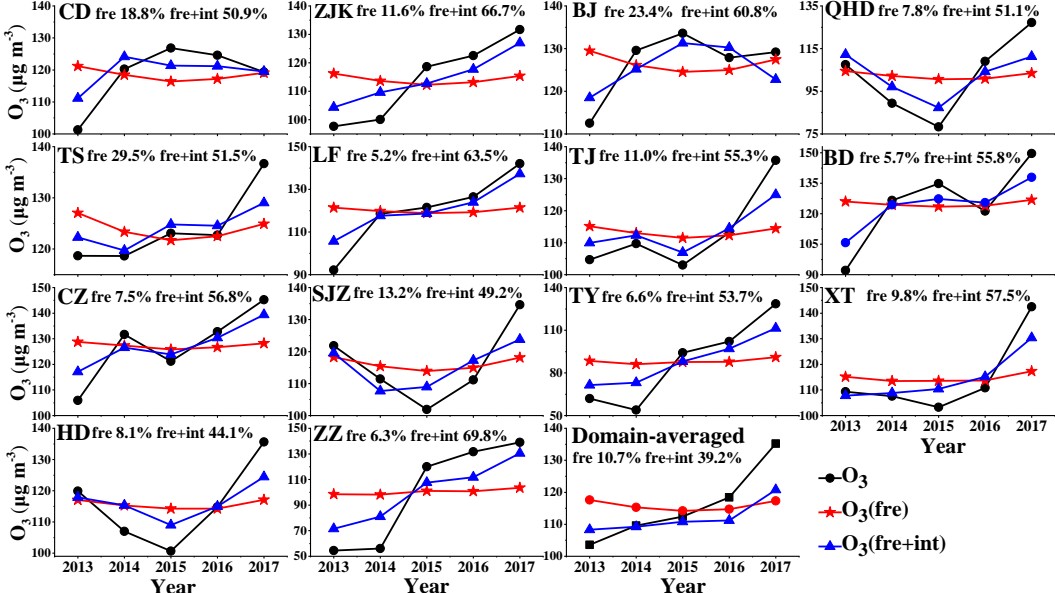

678

Fig. 8. The **interannual** MDA8 $O_3$ concentration trends for observed and reconstructed $O_3$ based on variations in weather types in 14 cities. The black lines represent the observed **interannual** MDA8 $O_3$ trend, whereas the red and blue lines are the trends of reconstructed MDA8 $O_3$ concentrations according to the frequency-only and both frequency and intensity of weather types changes, respectively. The percentages in each city indicate the $O_3$ **interannual** variabilities influenced by frequency-only and by both frequency and intensity of weather type changes.

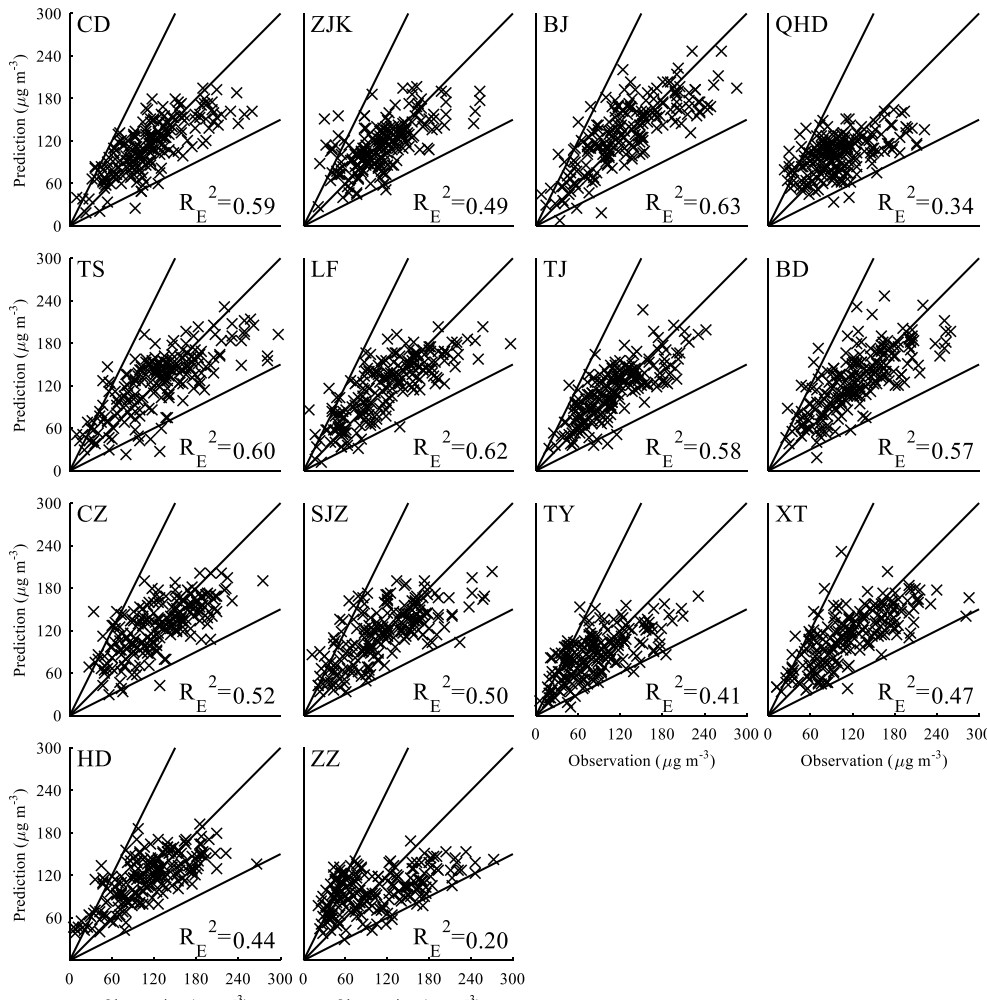

**Fig. 9. Scatter plots of predicted versus observed MDA8 O₃ concentrations for each city.** **The predicted**
**concentrations were obtained by inputting the validation data (20% of the total data) into the corresponding**
**model equations for five weather categories.** **The $R_E^2$ values indicate the percentage of explained variance in**
**the composite model that contains the building and validation datasets for each city. The three black lines**
**indicate 2:1, 1:1 and 1:2 ratio lines of predictions and observations.**