# Peer review of "Quantifying the impact of synoptic circulation on ozone variations in North China from April-October 2013-2017"

_Atmospheric Chemistry and Physics, 2019_

## Referee Comment (RC1) · Anonymous Referee #1 · 25 Jun 2019

Comment to acp-2019-485:

This manuscript titled "Quantifying the impact of synoptic circulations on ozone variations in North China from April–October 2013–2017" tried to find out the impacts of synoptic circulations and build a potential forecast model. I found several major problems needed to be addressed before publication in ACP.

Major comments:

1. Now that the grid of Era-interim is fine, thus why only 16-point grid were selected to classify the circulation. Furthermore, if there are some scientific considerations, the authors should clearly illustrate and connected to the $O_3$ pollution.

2. Closely to comment 1, the synoptic circulation classification must be connected to the features of MDA8 O3. In the manuscript, firstly 26 types were separated, and then 5 weather categories were summarized. I think better solution is to consider the MDA8 O3 in the first step. In other words, in the authors' scheme, the 26 types might be already diagnosed by many previous studies, even not related to the surface O3 pollution.

3. Only the sea level pressure was considered when synoptic circulation classification, it is better to add some variables in the mid-high troposphere. As you know, the atmospheric circulations in the mid-high

troposphere were more representativeness.

4.  The authors illustrated that "39.2% of the inter-annual domain-averaged O3 increase from 2013 to 2017 was 28 attributed to synoptic changes". I wonder how to discuss the interannual variations only using 5 years data.

5.  To provide the potential of O3 forecast, several models were built for each city and the results were shown in Section 3.4.

    (1)  How can we pre-determine the type of the synoptic circulations? One possible way is to use the output of numerical weather model.

    (2) The selected predictors in the models were the simultaneous variables, thus the question is how to obtain the predictors? If the answer is the output of NWP, the models should be trained from the achieved NWP output data instead of the observation or reanalysis.

    (3) TCC is not a routine observed variable, and also not a reliable NWP output.

6.  The English were intensively suggested to be improved by the native speaker.

Mini comments

1.  Line 24: what is the S-W-N stand for? cyclone type (C)….

    The use of abbreviations should be modified throughout the manuscript,

particularly in the Abstract.

2. What is the mean of QHD, ZZ?

3. Line 101-113: the type set is different.

4. Line 164: Why mentioned Figure 7a before Figure 2? Similar problems can be find in the manuscript.

5. Line 182: the definition of "exceedance ratio"?

6. Line 188: the reason for these 14 cities? That is, why the authors choose these 14 cities?

7. Line 219: " and and and" is not a good section title.

8. Figure 4 & 6: the panels are too small to read.

9. Figure 3: the color bae cannot show the 26 types.

---

## Referee Comment (RC2) · Anonymous Referee #2 · 9 Aug 2019

General Comments: This paper provides a thorough analysis of the synoptic circulation patterns that influence ozone variability across North China. The methods are based on previous work and are clearly stated. The ozone data are from the extensive Chinese monitoring network and provide a clear view of ozone's distribution and variability across the study region. The paper concludes with an estimate of the impact of weather patterns on ozone changes from 2013 to 2017, which is an important and useful result. The paper could be acceptable for publication in ACP after the authors address my comments and concerns. My two main concerns deal with the need for improved referencing of previous work, and more details and motivation are needed regarding the ozone forecast equations. The standard of English is quite good, but there

are many instances of grammatical errors or awkward phrasing. The paper should be revised for English prior to publication.

Major Comments:

The title should be modified so that is uses the term "circulation pattern" rather than "circulations". Circulation pattern is defined by the Glossary of the American Meteorological society, while circulations is not defined: http://glossary.ametsoc.org/wiki/Circulation_pattern Also, variability is a better word choice than variations. Therefore the title should be: "Quantifying the impact of synoptic circulation patterns on ozone variability in North China from April-October, 2013-2017" Throughout the paper "synoptic circulation pattern" should be the preferred term

The Introduction does a very poor job of referencing relevant papers in support of the claims made in the text: 1) Knowland et al 2017 is a nice paper on transport patterns, but it is not an appropriate reference regarding the impact of ozone on human health and vegetation. Likewise, Jacob and Winner focuses on how climate change will affect ozone. Instead see Fleming et al. [2018] and Mills et al. [2018]. 2) the papers by Liu et al [2007; 2012] focus on PM2.5 and are not authoritative papers for explaining ozone photochemistry. Instead see Monks et al [2009; 2015]. 3) Line 51: please provide a reference for APAPPC 4) Line 58: references are need for the impact of climate on ozone. Jacob and Winner is appropriate, as is Lu et al. [2019]. 5) Regarding the background on the association between transport patterns and air pollution, this paper is missing some key references (listed below), such as Moody et al. [1998] and Cooper et al. [2001]. Also, Section 4.4 of Monks et al. [2009] provides a very good summary of the early work (through 2008) on the relationship between synoptic patterns and air pollution transport. An important paper for China is Wang et al., 2009.

I have many questions regarding Section 3.4: 1) The authors need to explain why they developed the equations to forecast ozone. Is this just an academic exercise to see if its's even possible? Has this method been requested by air quality managers? Is it

an alternative to atmospheric chemistry models that have not performed well? 2) How would this method work in operational mode? Would a weather forecast model be run to identify the synoptic circulation pattern for the next day, and then for a given city, select the equation that predicts ozone for that particular synoptic circulation pattern? Does Figure 9 show results for each city, using all five major synoptic circulation patterns? 3) The authors provided some summary statistics in the Supplement (Table S4) to describe the performance of the forecast equations, but these results are not very intuitive or easy to understand. It would really help if the authors can select a typical city and then report the predicted ozone values for the five major synoptic circulation patterns, and then report the range of values that were actually observed. For example, if the model predicts Beijing will have ozone of 160 ug m-3 tomorrow under Cyclonic conditions, but the observations show a range of 140 +/- 50 ug m-3 (where the uncertainty is 2 standard deviations) , the reader might conclude that the while the equation predicts a high ozone day there is a wide range of uncertainty. 4) Are the forecasts from these equations any better than the forecasts from an atmospheric chemistry model?

Minor Comments:

Line 63-65 This sentence is not well written. A better form would be: A given synoptic circulation pattern represents a particular range of meteorological conditions, therefore synoptic classification is a useful method for gaining insight into the impact of meteorology on ozone levels on the regional scale.

Line 182 Please explain how the exceedance ratio is calculated

Line 186 Beijing is a city, is it not? Or is "Beijing" referring to a large urban region that contains smaller cities?

Line 208 These references are not authoritative when describing the impacts of stratospheric ozone on China. A good overview is provided by Stohl et al., and a recent model analysis that quantifies the impact of the stratosphere on ozone above China is Verstraeten et al., 2015.

Line 209 It's not clear to me how the results of Tang et al. differ from your results. Please clearly state the results from Tang et al. and then show how your results are different.

Line 216 Not all of these references are authoritative. A good review of the relationship between ozone and temperature is Pusede et al., 2015.

Lines 256-259 These ozone fluctuations in relation to cold fronts have been reported for the eastern USA: Cooper et al., 2001 and Cooper et al., 2002

Figure 1. The blue boxes around some of the cities are very difficult to see. Please make the lines thicker.

Figure 4. The panels in this figure are far too small to be seen. This figure needs to fill an entire page, so that the reader can clearly see the information. Likewise, Figure S8 is impossible to read.

References:

Cooper, O. R., J. L. Moody, D. D. Parrish, M. Trainer, J. S. Holloway, T. B. Ryerson, G. Hübler, F. C. Fehsenfeld, S. J. Oltmans and M. J. Evans (2001), Trace gas signatures of the airstreams within North Atlantic cyclones - Case studies from the NARE'97 aircraft intensive, J. Geophys. Res., 106, 5437-5456, doi:10.1029/2000JD900574.

Cooper, O. R., J. L. Moody, D. D. Parrish, M. Trainer, J. S. Holloway, G. Hübler, F. C. Fehsenfeld, and A. Stohl (2002), Trace gas composition of midlatitude cyclones over the western North Atlantic Ocean: A seasonal comparison of O3 and CO, J. Geophys. Res., 107(D7), 4057, doi:10.1029/2001JD000902.

Fleming, Z. L., et al. (2018), Tropospheric Ozone Assessment Report: Present-day ozone distribution and trends relevant to human health, Elem Sci Anth, 6(1):12, DOI: https://doi.org/10.1525/elementa.273

Lu, X., et al., 2019. Surface and tropospheric ozone trends in the Southern Hemi-

sphere since 1990: possible linkages to poleward expansion of the Hadley Circulation. Science Bulletin, 64(6), pp.400-409.

Mills, G., et al. (2018), Tropospheric Ozone Assessment Report: Present-day tropospheric ozone distribution and trends relevant to vegetation, Elem. Sci. Anth., 6(1):47, DOI: https://doi.org/10.1525/elementa.302

Monks, P. S., et al. (2015), Tropospheric ozone and its precursors from the urban to the global scale from air quality to short-lived climate forcer, Atmos. Chem. Phys., 15, 8889-8973, doi:10.5194/acp-15-8889-2015

Monks, P. S., et al. (2009), Atmospheric Composition Change – Global and Regional Air Quality, Atmos. Environ., 43, 5268-5350.

Moody, J. L., J. W. Munger, A. H. Goldstein, D. J. Jacob, and S. C. Wofsy (1998), Harvard Forest regional-scale air mass composition by Patterns in Atmospheric Transport History (PATH), J. Geophys. Res., 103, 13,181–13,194.

Pusede SE, Steiner AL and Cohen RC (2015). Temperature and Recent Trends in the Chemistry of Continental Surface Ozone. Chemical Reviews 115(10): 3898-3918, doi:10.1021/cr5006815.

Verstraeten, W. W., Neu, J. L., Williams, J. E., Bowman, K. W., Worden, J. R., and Boersma, K. F.: Rapid increases in tropospheric ozone production and export from China, Nat. Geosci., 8, 690–695, https://doi.org/10.1038/ngeo2493, 2015. 

Wang, T.,Wei, X.L., Ding, A.J., Poon, C.N., Lam, K.S., Li, Y.S., et al., 2009a. Increasing surface ozone concentrations in the background atmosphere of Southern China, 1994–2007. Atmos. Chem. Phys. 9, 6217–6227.

---

## Author Response (AR1)

**Responses to the reviewers' comments point by point**

We thank the reviewer for their comments, and we do think their comments and suggestions improved our manuscript considerably. Our point-by-point replies to the comments are given below. The replies are in blue font, the revisions in the revised manuscript and the responses with red font, and important statements are of bold.

**Reviewer1:**

**Major comments:**

1. Now that the grid of Era-interim is fine, thus why only 16-point grid were selected to classify the circulation. Furthermore, if there are some scientific considerations, the authors should clearly illustrate and connected to the $O_3$ pollution.

Reply:

Thanks for the comment. The description of the method was not sufficient, so we have added more details about the reason for using the Lamb-Jenkinson weather type scheme and the description of this method in the revised paper and Text S1 (lines 21-76 of supplementary material).

In synoptic climatology, along with subjective or manual approaches, objective or automated approaches are widely used synoptic weather typing procedures to identify recurring map patterns or variable groups that typify significant modes of circulation and to classify each case into one of these modes (Yarnal, 1993; Huth et al., 2008). There are many objective methods, such as correlation-based map-pattern technique, sums-of-squares method, rotated principal component analysis, hierarchical clustering (average linkage or Ward's clustering), and K-means clustering. As suggested by Yarnal (1993) and Huth (1996), all the methods proved to be capable of yielding meaningful classifications and none of them can be thought of as the best in all aspects. Which method to use will depend mainly on the aim of the classification. Notably, the final number of synoptic types using these algorithms is associated with a given period

and region, statistical algorithms, prior knowledge of the synoptic climatology of the region, and experimentation with various statistical procedures; finally, a subjective decision as to how many clusters are appropriate for the study period is made by investigator. Thus, the results of a synoptic-type analysis are quite subjective.

**The noted British climatologist Lamb developed a synoptic-scale, daily weather-map classification for use over the British Isles, and seven basic types were identified manually (Lamb, 1972). . Based on Lamb's study, Jenkinson (1977) improved the subjective approach to an objective approach, called the Lamb-Jenkinson method (Jones et al., 1993; Trigo and Dacamara, 2000). According to the sea level pressure (SLP) of these 16 grids, a set of indices related to the direction and vorticity of geostrophic flow are calculated to determine the weather type. The indices used are the following: southerly flow component of the geostrophic surface wind (SF), westerly flow component of the geostrophic surface wind (WF), resultant flow (FF), southerly shear vorticity (ZS), westerly shear vorticity (ZW) and total shear vorticity (Z). These indices were computed using SLP values obtained for the retained number of grid points and are both for the flow units and for the geostrophic vorticity expressed in hPa. As shown in 1a, the research area is located in the central position, which refers to the area connecting with P4, P8, P12, P13, P9 and P5. The SLP of 16 grids can be used to characterize the distance of between the study region and high-/low-pressure system; therefore, the method is available to classify the weather pattern for each day and has been successfully applied in many areas** (Lamb, 1972; Jenkinson, 1977; Trigo and Dacamara, 2000; Demuzere et al., 2009; Santurtún et al., 2015; Pope et al., 2016; Liao et al., 2017). The following presents the calculation methods for each index:

SF=1.035×[0.25×(P5+2×P9+P13)-0.25×(P4+2×P8+P12)]

WF=[0.5×(P12+P13)-0.5×(P4+P5)]

ZS=0.85×[0.25(P6+2×P10+P14)-0.25×(P5+2×P9+P13)-
0.25×(P4+2×P8+P12)+0.25×(P3+2×P7+P11)]

ZW=1.12×[0.5×(P15+P16)-0.5×(P8+P9)]-0.91×[0.5×(P8+P9)-0.5×(P1+P2)]

F=(SF2+WF2)1/2

$$Z = ZS + ZW$$

[Figure]

P represents the SLP at the grid point. The positions of 16 grid points are shown in Fig. 1a; for example, P1 is the SLP at the 1st grid point.

The weather types are defined by comparing values of FF and Z:

(1) Direction of flow (in degrees) is given by $\tan^{-1}(WF/SF)$, 180° being added if WF is positive. The appropriate wind direction is computed using an eight-point compass, allowing 45° per sector.

(2) If $|Z| < FF$, flow is essentially straight and considered to be of a pure directional type (eight different possibilities according to the compass directions).

(3) If $|Z| > 2FF$, the pattern is considered to be of a pure cyclonic type if $Z > 0$ or of a pure anticyclonic type if $Z < 0$.

(4) If $FF < |Z| < 2FF$, flow is considered to be of a hybrid type and is therefore characterized by both direction and circulation (16 different types).

Thus, compared with other objective synoptic classification approaches, the advantage of Lamb-Jenkinson method is that the number of synoptic types and the weather type that is present each day in the specific region is robust and fixed. In addition, the method clearly gives the typical pressure fields (anticyclone, cyclone, directional types and hybrid types), which can well reflect the wind fields over the study region. Particularly, directional types can represent the prevailing wind direction in this area under the control of the specific weather pattern. Many studies have shown that the high/low concentrations of ozone are always associated with the southerly/northerly winds in North China (Han et al., 2019; Li et al., 2019) . Consequently, the Lamb-

Jenkinson weather type scheme is a better method for exploring the O$_3$ pollution in North China.

2. Closely to comment 1, the synoptic circulation classification must be connected to the features of MDA8 O$_3$. In the manuscript, firstly 26 types were separated, and then 5 weather categories were summarized. I think better solution is to consider the MDA8 O$_3$ in the first step. In other words, in the authors' scheme, the 26 types might be already diagnosed by many previous studies, even not related to the surface O$_3$ pollution.

Reply:

**There are two ways to discuss the relationship between synoptic circulation patterns and air pollution: the environmental-to-circulation approach** and **the circulation-to-environmental approach (Yarnal, 1993; Demuzere et al., 2009).**

For the **environmental-to-circulation** approach, the circulation data are based on the criteria defined by the environmental variable (e.g., O$_3$), so it can be of use in a descriptive way to obtain more insight in those patterns involved in regulating the magnitude of surface environmental variables. However, unlike circulation-to-environmental approach, it lacks any capability to quantify the impact of meteorological factors on air pollutant and prediction. Conversely, **the circulation-to-environmental approach classifies the circulation patterns based on standard pressure fields (e.g., SLP or 500 hPa geopotential height) prior to seeking the links between the environmental variable and the circulation data. This approach follows the hypothesis that circulation conditions have a distinctive effect on a certain environmental variable.** In this study, we adopt the latter. **Firstly, circulation classification can typically represent the complete range of the atmospheric circulation over the area and the entire period for which data are available. In addition, the classification of circulation data is independent of the environmental response (Yarnal, 1993). Therefore, it has been widely used for discussing the relationship between synoptic weather patterns and atmospheric pollutants, such as ozone, PM$_{2.5}$, and PM$_{10}$ (Demuzere et al., 2009; Demuzere and van Lipzig, 2010a; Santurtún et al., 2015; Pope et al., 2016; Liao et al., 2017).** Above all, the approach

provides a basis for quantifying the relationship between $O_3$ concentration and different circulation patterns and reconstructing and predicting the $O_3$ concentration caused by synoptic and local meteorological influence, **allowing the effects of the weather type changes on the inter-annual and day-to-day ozone variability to be evaluated.**

The weather types are developed using Era-interim mean sea level pressure data, and for a given day, they describe the location of high- and low-pressure centers that determine the direction of the geostrophic flow. First, the relations between synoptic patterns and $O_3$ concentration vary over different regions. For example, anticyclonic conditions and easterly flows have been shown to significantly enhance ozone concentrations over the UK in summer (Pope et al., 2016) , but in Spain, the median concentrations were statistically significantly lower on days with anticyclonic weather conditions than in the rest of meteorological situations, with the maximum values found on days with northern and eastern components (Querol et al., 2014). In addition, **due to the differences in the topography, pollution source, local circulation, etc., the relations between these factors and $O_3$ concentration vary over different regions as well. Demuzere et al. (2009) demonstrated higher surface $O_3$ concentrations in summer in an easterly weather type at a rural site in Cabauw, Netherlands, whereas an opposite result was obtained by Liao et al. (2017) in the Yangtze River Delta region in eastern China. Therefore, Lamb-Jenkinson synoptic classification and its relationship with O3 needs to be explored separately in different regions, especially in North China.**

3. Only the sea level pressure was considered when synoptic circulation classification, it is better to add some variables in the mid-high troposphere. As you know, the atmospheric circulations in the mid-high troposphere were more representativeness.
Reply:
Thanks for your suggestion. **A method to classify daily circulation patterns was originally developed by Lamb (1972), which is a subjective classification method. The method used surface pressure synoptic charts describing the flow in the 500-hPa level in the atmosphere.** To avoid dependency of the daily weather maps on

experience and consistency of the researcher, this method was objectified by Jenkinson (1977); as a result, this method has been upgraded to objective classification method. Moreover, as shown by Conway and Jones (1998), **circulation patterns fundamentally control meteorological characteristics on the surface, whereby the use of surface level pressure has several advantages.** The study done by Mckendry et al. (2006) showed that **upper pressure level patterns are less variable than surface pressure patterns and that particular upper level patterns may be associated with a large range of sea-level pressure synoptic types.** The surface pressure field can better represent the local meteorological factors. Therefore, sea level pressure is more appropriate for the classification of circulation patterns (Demuzere et al., 2010).

4. The authors illustrated that "39.2% of the inter-annual domain-averaged O3 increase from 2013 to 2017 was 28 attributed to synoptic changes". I wonder how to discuss the interannual variations only using 5 years data.

Reply:

The quantifying work was reported by Hegarty et al. (2007). She indicated that 46% of the inter-annual variability in summertime $O_3$ was caused by synoptic changes with intensity being the dominant factor based on 5 years (2000-2004) of ozone data in the northeastern USA. The basic principle of the method is to find out the relevance of the change in the synoptic intensity to ozone inter-annual variability. We upgraded this method by using six intensity factors that reflect the changes in the synoptic intensity and found the strongest correlation intensity index with ΔO3 under each weather type in each city. Then, we reconstructed the inter-annual ozone levels by taking into account either frequency-only or both frequency and intensity variations of synoptic circulations as introduced in Section 2.4. After reconstructing the $O_3$ concentration ($\overline{O_{3\,m}}$(fre + int)), $\Delta\overline{O_{3m}}$(fre + int) is the difference between maximum and minimum annual reconstructed ozone values by considering the effects of both frequency and intensity of synoptic weather patterns; $\Delta O_3$ \_obs indicated the maximum and minimum difference of annual observed ozone concentration. Therefore, $\mathbf{\Delta\overline{O_{3m}}(fre + int)/}$

**$\Delta O_3\_obs$ indicates the inter-annual oscillations in ozone levels caused by synoptic variability, introduced in Section 3.3.2. The ratio of the oscillation of ozone concentration caused by meteorological factors to the oscillation of actual ozone concentration is the impact of meteorological conditions on the interannual variation of ozone concentration.** In this paper, inter-annual variability in domain-averaged observed MDA8 $O_3$ in 14 cities varied from averaged maximum values of 135 μg m$^{-3}$ in 2017 to a minimum 104 μg m$^{-3}$ in 2013. The contributions of circulation patterns variations in inter-annual $O_3$ increase was 39.2%, and the remaining inter-annual variability was possibly due to nonlinear relationships with recent emission control measures over North China. Therefore, the five spots (years) are sufficient to illustrate the inter-annual variations.

5. To provide the potential of O3 forecast, several models were built for each city and the results were shown in Section 3.4.

(1) How can we pre-determine the type of the synoptic circulations?

One possible way is to use the output of numerical weather model.

Reply:

Yes, we intend to use the forecasting SLP data from numerical weather model (e.g., WRF) to determine day-to-day weather patterns.

(2) The selected predictors in the models were the simultaneous variables, thus the question is how to obtain the predictors? If the answer is the output of NWP, the models should be trained from the achieved NWP output data instead of the observation or reanalysis.

Reply:

**First, the relevance of establishing this equation is to quantify the effect of synoptic changes in weather patterns on day-to-day ozone concentration and then to establish the ozone potential forecast model.**

**To better reflect this view, we added the following sentences in lines 86-89 of the revised manuscript: 'Quantifying the contribution of local meteorological factors**

**to the day-to-day variation in ozone will a provide scientific basis and guidance for reasonable ozone reduction measures, and clarifying and quantifying the relationship between meteorological factors and ozone is vital for daily ozone pollution potential forecasts.'**

In order to accurately express the relationship between the actual local meteorological factors and ozone concentration in the atmosphere, we input the measured meteorological factors for building and validating the model. As for the prediction stage, we believe that the meteorological factors simulated by numerical models are credible for the short-term forecasting.

(3) TCC is not a routine observed variable, and also not a reliable NWP output.

Reply:

Thank you very much for your valuable comments. We initially considered the meteorological variables simulated by NWP. TCC can be obtained indirectly from the model; however, considering its complexity and its influence on ozone in the prediction model, **we do not consider TCC anymore and rebuilt the ozone potential prediction model. When TCC is excluded, it has little effect on the results of the model. The corrected results are shown in Section 3.4, Table 2 and Tables S3-S4.**

The following sentences are the result of the revision shown in lines 37-39, 397-398 and 431-433 of the revised manuscript, respectively.

'Overall, 41-63% of the day-to-day variability of MDA8 $O_3$ concentrations was due to local meteorological variations in most cities over North China, except for two cities: QHD (Qinhuangdao) at 34% and ZZ (Zhengzhou) at 20%.'

'The result of validation shows that $R^2$ was higher than 0.50 except for QHD, SJZ and ZZ (0.24-0.47), while CV was lower than 40% except for TY and ZZ.'

'Local meteorological parameters could explain 57-63% and 41-52% of the day-to-day MDA8 $O_3$ concentration variability for the northern cities (except for QHD, 34%) and southern cities (except for ZZ, 20%), respectively.'

6. The English were intensively suggested to be improved by the native speaker.

Reply:

Thanks; we have revised the language problems with the help of native speaker in this revised version. Language modification is not marked in the revised manuscript in red font.

**Minor comments:**

1. Line 24: what is the S-W-N stand for? cyclone type (C)….
The use of abbreviations should be modified throughout the manuscript, particularly in the Abstract.

Reply:

Thanks for the comment. The revised sentence, as shown in lines 27-29, reads as follows: 'S-W-N directions (geostrophic wind direction diverts from south to north), LP (low-pressure related weather types) and C (cyclone type, controlled by low pressure center)'

2. What is the mean of QHD, ZZ?

Reply:

We have added the description to the article. As shown in lines 37-39 of revised manuscript, the revised sentence is as follows:

'Overall, 41-63% of the day-to-day variability of MDA8 $O_3$ concentrations was due to local meteorological variations in most cities over North China, except for two cities: QHD **(Qinhuangdao)** at 34% and ZZ **(Zhengzhou)** at 20%.'

3. Line 101-113: the type set is different.

Reply:

Thanks for your suggestion, as shown in lines 111-118 of revised manuscript, the revised sentences are as follows:

"Each city has at least two monitoring sites, and the city MDA8 $O_3$ levels are the corresponding averages over all sites in that city. MDA8 $O_3$ values were collected in

only 14 cities for the time period 2013-2017 and in an additional 44 cities for the time period 2015-2017, with detailed information shown in Fig. 1 and Table S1. The original unit of the ozone observations is µg m$^{-3}$, and the converted coefficient from mixing ratios (unit: ppbv) to µg m$^{-3}$ is a constant (e.g., 0.5 at temperature of 25 °C and pressure of 1013.25 hPa). In this study, we will use the original unit. Unless otherwise noted, **the analysis of O$_3$ refers to MDA8 O$_3$** during April-October in this paper."

4. Line 164: Why mentioned Figure 7a before Figure 2? Similar problems can be find in the manuscript.

Reply:

We have tried to set the figures in order, but when the complex method is introduced in Section 2, the figure is also needed. Thus, we have to adjust the order based on the main content.

5. Line 182: the definition of "exceedance ratio"?

Reply:

Thanks a lot. We ignored the definition of 'exceedance ratio' in our paper. We added the sentence as shown in lines 193-194 of revised manuscript, as follows: exceedance ratio which means the proportion of days exceeding the standard (160 µg m$^{-3}$).

6. Line 188: the reason for these 14 cities? That is, why the authors choose these 14 cities?

Reply:

These 14 cities first started monitoring in North China in 2013, and they have 5-year ozone data; additionally, they can represent the pollution situation in North China to a great extent.

7. Line 219: "and and and" is not a good section title.

Reply:

We have revised the section title as 'The meteorological conditions and regional ozone concentrations under different predominant circulation type'.

8. Figure 4 & 6: the panels are too small to read.

Reply:

We have revised the unclear figures in the article, as shown below:

Figure 4:

[Figure]

**Fig. 4. Mean surface pressure field (unit: hPa) for the 26 weather types during April-October of 2013-2017 and occurrence days (1070 days in total). '*' indicates the center of North China.**

Figure 6:

[Figure]

**Fig. 6. Spatial distribution of average MDA8 O₃ for the 26 weather types. The first, second, and third rows correspond to the weather categories N-E-S direction, S-W-N direction and LP, respectively, and the fourth row includes both categories C and A.**

9.Figure 3: the color bae cannot show the 26 types.

Reply:

We have revised the unclear figure in the article, as shown below:

Figure 3:

[Figure]

**Fig. 3. Interannual (a) and monthly (b) averaged concentrations of ozone and frequencies of 26 weather types from April-October 2013-2017. The red dots represent the mean values, the vertical red lines indicate the standard deviations, and stacked charts represent the percentages of various weather types (2013 and 2014 are averaged for 14 cities; 2015-2017 are averaged for 58 cities). The pink, orange, light blue, dark blue and black areas represent the weather categories N-E-S direction, S-W-N direction, LP, C and A, respectively.**

**Reviewer2:**

**General Comments:**

My two main concerns deal with the need for improved referencing of previous work, and more details and motivation are needed regarding the ozone forecast equations. The standard of English is quite good, but there are many instances of grammatical errors or awkward phrasing. The paper should be revised for English prior to publication.

Reply:

Thanks for the comment. Referring to your comments, we have reconsidered the references in the article; revised the problems in the section of the ozone forecast equations and the language problems with the help of native speaker. Language modification is not marked in the revised manuscript in red font.

**Major comments:**

1. The title should be modified so that is uses the term "circulation pattern" rather than "circulations". Circulation pattern is defined by the Glossary of the American Meteorological society, while circulations is not defined: http://glossary.ametsoc.org/wiki/Circulation_pattern

Also, variability is a better word choice than variations. Therefore, the title should be: "Quantifying the impact of synoptic circulation patterns on ozone variability in North China from April-October, 2013-2017" Throughout the paper "synoptic circulation pattern" should be the preferred term.

Reply:

Thanks for the comment. **We reconsidered the terms 'circulation pattern' and 'weather type' and revised them in our paper. Based on previous references, we think the best term to describe the Lamb weather typing technique in the entire revised manuscript is 'weather types,' and in other instances, we used 'circulation pattern.' we also replaced 'variation' with 'variability.'.**

2. The Introduction does a very poor job of referencing relevant papers in support of the claims made in the text:

2) the papers by Liu et al [2007; 2012] focus on PM2.5 and are not authoritative papers for explaining ozone photochemistry. Instead see Monks et al [2009; 2015].

3) Line 51: please provide a reference for APAPPC

4) Line 58: references are need for the impact of climate on ozone. Jacob and Winner is appropriate, as is Lu et al. [2019].

5) Regarding the background on the association between transport patterns and air pollution, this paper is missing some key references (listed below), such as Moody et al. [1998] and Cooper et al. [2001]. Also, Section 4.4 of Monks et al. [2009] provides a very good summary of the early work (through 2008) on the relationship between synoptic patterns and air pollution transport. An important paper for China is Wang et al., 2009.

Reply:

Thank you very much for your valuable suggestion. According to your suggestion, we revised the references in this article.

On comment 3), We revised as shown in lines 55-56, as follows: 'the Action Plan for Air Pollution Prevention and Control (www.gov.cn/zwgk/2013-09/12/content_2486773.htm) was implemented.'

3. I have many questions regarding Section 3.4:

1) The authors need to explain why they developed the equations to forecast ozone. Is this just an academic exercise to see if its's even possible? Has this method been requested by air quality managers? Is it an alternative to atmospheric chemistry models that have not performed well?

Reply:

In this section, we did not express our meaning clearly. The reason for establishing this equation is to quantify the effect of synoptic weather patterns changes on day-to-day ozone concentration and then to establish the ozone potential forecast model. Therefore,

we changed the title of 3.4 as shown in lines 363-364, as follows: '**Quantifying the impact of weather patterns on day-to-day ozone concentration and forecasting daily ozone concentration.**'

**The primary motivation of this study is to provide a comprehensive and quantitative understanding of how weather influences ozone pollution in Northern China; thus, we aim to quantify the impacts of synoptic weather patterns and local meteorological factors on daily variations of surface ozone in Northern China. We also want to search for a linkage between the daily variation of surface ozone and the local and synoptic meteorological factors statistically.** To better reflect this view, we added the following sentences in lines 86-89 of the revised manuscript: '**Quantifying the contribution of local meteorological factors to the day-to-day variation in ozone will a provide scientific basis and guidance for reasonable ozone reduction measures, and clarifying and quantifying the relationship between meteorological factors and ozone is vital for daily ozone pollution potential forecasts.**'

**Multiple linear regression (MLR) is an effective and widely used way to describe the relationship between meteorology and air quality and thus to aid in the prediction of air quality (Shen et al., 2015; Otero et al., 2016; Li et al., 2019). Compared to atmospheric chemistry models, the potential forecasting model of ozone is reliable and requires a low computational burden, which means the potential ozone could be assessed quickly and accurately in the short term or even in the medium/long term against the background of global climate change, which has been done in some regions (Cheng et al., 2007; Demuzere and van Lipzig, 2010b).**

2) How would this method work in operational mode? Would a weather forecast model be run to identify the synoptic circulation pattern for the next day, and then for a given city, select the equation that predicts ozone for that particular synoptic circulation pattern? Does Figure 9 show results for each city, using all five major synoptic circulation patterns?

Reply:

We intend to use the results of numerical weather model (e.g., WRF) to determine day-to-day weather patterns. **Since the Lamb weather classification method is not limited by the sample size, the weather classification results can be obtained as long as there is a predicted sea level pressure field. Therefore, we can easily determine the predicted weather type.** Other meteorological factors also depend on the results of the numerical weather model. Referring to the views of Reviewer 1, we reconsidered that TCC can be obtained indirectly from the model; however, considering the complexity of indirect computation and its influence on ozone in the prediction model, we do not consider TCC anymore and rebuilt the ozone potential prediction models in 14 cities.

We realized that we did not emphasize the percentage of the modeled data and validated data in Section 2.5. We added the following description (also as shown in lines 180-182 of the revised manuscript): **'Notably, in this research, after excluding the missing data and disordering the time sequences, 80% of these days were used to build the potential forecast equations and the remaining 20% were used to validate the accuracy of the equations.' Therefore, Figure 9 shows the validation of the remaining 20% of the data (not input into the model) by the modeling equation for each city. For validation data, the prediction of ozone concentration obtained by inputting the meteorological factors into the corresponding weather category's simulated formula of specific city; therefore, composite validation datasets indicate the prediction ozone concentrations of five categories are integrated (as shown in lines 394-397).**

3) The authors provided some summary statistics in the Supplement (Table S4) to describe the performance of the forecast equations, but these results are not very intuitive or easy to understand. It would really help if the authors can select a typical city and then report the predicted ozone values for the five major synoptic circulation patterns, and then report the range of values that were actually observed. For example, if the model predicts Beijing will have ozone of 160 ug m-3 tomorrow under Cyclonic

conditions, but the observations show a range of 140 +/- 50 ug m-3 (where the uncertainty is 2 standard deviations) , the reader might conclude that the while the equation predicts a high ozone day there is a wide range of uncertainty.

Reply:

In order to better illustrate the accuracy of the predicted results, we added in the supplement a comparison figure between the simulated results and the measured results from April to October 2013-2017. The added description is shown in the fourth paragraph of Section 3.4.

The results reveal that most of the validation data are within the acceptable error range within the 2:1 and 1:2 ratio lines, and the scatters are distributed evenly around the 1:1 line. **For example, the comparison of the observed and predicted ozone in Beijing during our study period is shown in Fig. S10. This also indicates that the segmented synoptic-regression approach is practicable to construct the ozone potential forecasting model in most cities in North China.** (as shown in lines 400-403)

[Figure]

Fig. S10. Comparison of the observed and predicted values in Beijing from April-October 2013-2017

4) Are the forecasts from these equations any better than the forecasts from an atmospheric chemistry model?

Reply:

The MLR method used in this study is based on the synoptic weather patterns; thus, it is called as a segmented synoptic-regression analysis approach. **This method is superior to the traditional simulation only using local meteorological factors and can more accurately quantify the impact of meteorological conditions on day-to-day ozone concentration (Eder et al., 1994; Demuzere and van Lipzig, 2010a).**

The prediction model of ozone concentration established by the synoptic-regression method is a potential prediction model based on meteorological factors. **Its function is not to accurately predict the concentration of ozone but to predict the possible concentration of ozone under certain weather conditions in the future. This method is simple and rapid in assessing future ozone concentrations for short-range or even mid- and long-term forecasting.**

**Minor Comments:**

Line 63-65 This sentence is not well written. A better form would be: A given synoptic circulation pattern represents a particular range of meteorological conditions, therefore synoptic classification is a useful method for gaining insight into the impact of meteorology on ozone levels on the regional scale.

Reply:

Thank you. We have replaced this statement with the previous description, as shown in lines 65-67.

Line 182 Please explain how the exceedance ratio is calculated

Reply:

Thank you. We added the sentence as shown in lines 193-194 of revised manuscript, as follows: 'exceedance ratio which means the proportion of days exceeding the standard (160 μg m$^{-3}$).'

Line 186 Beijing is a city, is it not? Or is "Beijing" referring to a large urban region that contains smaller cities?

Reply:

Beijing refers to an administrative division.

Line 208 These references are not authoritative when describing the impacts of stratospheric ozone on China. A good overview is provided by Stohl et al., and a recent model analysis that quantifies the impact of the stratosphere on ozone above China is Verstraeten et al., 2015.

Reply:

Thank you. We added these references as shown in line 221 of revised manuscript.

Line 209 It's not clear to me how the results of Tang et al. differ from your results. Please clearly state the results from Tang et al. and then show how your results are different.

Reply:

Thank you. We added the sentence as shown in lines 221-225 of revised manuscript, as follows: 'Notably, this conclusion is different from that of Tang et al. (2012) , ,who reported that the concentration in July was higher than that in May in North China during 2009-2010. However, as our study shows that the domain-averaged MDA8 O3 in May was even higher than in July, the concentrated pollution episode occurred earlier, especially in 2017.'

Line 216 Not all of these references are authoritative. A good review of the relationship between ozone and temperature is Pusede et al., 2015.

Reply:

Thank you. We added the reference as shown in line 230 of revised manuscript.

Lines 256-259 These ozone fluctuations in relation to cold fronts have been reported for the eastern USA: Cooper et al., 2001 and Cooper et al., 2002

Reply:

Thank you. We added the references as shown in line 274 of revised manuscript.

Figure 1. The blue boxes around some of the cities are very difficult to see. Please make the lines thicker.

Reply:

Thank you. We repainted the figure as follows:

[Figure]

Fig. 1. Location of North China (shaded area), all cities (black spots) and sea level pressure grids (a). The 16 red points show the locations of the 5°×10° mean sea level pressure grids used for the Lamb-Jenkinson weather type classification. The spatial distributions of the maximum daily 8-h running average O₃ (MDA8 O₃) concentration (b) and exceedance ratios (c) for 58 cities. Statistics for 2013-2017 are shown with blue boxes; the other boxes are those for 2015-2017. The base map is topography; the elevations of the Taihang Mountains are more than 1200 meters, and the Yan Mountains range from 600 to 1500 meters.

Figure 4. The panels in this figure are far too small to be seen. This figure needs to fill an entire page, so that the reader can clearly see the information. Likewise, Figure S8 is impossible to read.

Reply:

Thank you. In order to see the information of Fig. S8, we repaint Fig. S8 into two figures, Fig. S8 and Fig. S9, as shown below:

[Figure]

Fig. S8. Pressure field characteristics and occurrence days (n) of weather type N to CE in 2013-2017.

[Figure]

Fig. S9. Pressure field characteristics and occurrence days (n) of weather type CSE to ANW in 2013-2017.

**References**

Cheng, C., Campbell, M., Qian, L., Li, G., Auld, H., Day, N., Pengelly, D., Gingrich, S., and Yap, D.: A Synoptic Climatological Approach to Assess Climatic Impact on Air Quality in South-central Canada. Part II: Future Estimates, Water Air & Soil Pollution, 182, 117-130, 2007.

Conway, D., and Jones, P. D.: The use of weather types and air flow indices for GCM downscaling, Journal of Hydrology, 212-213, 348-361, https://doi.org/10.1016/S0022-1694(98)00216-9, 1998.

Demuzere, M., Trigo, R. M., Vila-Guerau de Arellano, J., and van Lipzig, N. P. M.: The impact of weather and atmospheric circulation on O3 and PM10 levels at a rural mid-latitude site, Atmos. Chem. Phys., 9, 2695-2714, https://doi.org/10.5194/acp-9-2695-2009, 2009.

Demuzere, M., and van Lipzig, N. P. M.: A new method to estimate air-quality levels using a synoptic-regression approach. Part I: Present-day O3 and PM10 analysis, Atmospheric Environment, 44, 1341-1355, https://doi.org/10.1016/j.atmosenv.2009.06.029, 2010a.

Demuzere, M., and van Lipzig, N. P. M.: A new method to estimate air-quality levels using a synoptic-regression approach. Part II: Future O3 concentrations, Atmospheric Environment, 44, 1356-1366, 10.1016/j.atmosenv.2009.06.019, 2010b.

Demuzere, M., Werner, M., Lipzig, N. P. M. V., and Roeckner, E.: An analysis of present and future ECHAM5 pressure fields using a classification of circulation patterns, International Journal of Climatology, 29, 1796-1810, 2010.

Eder, B. K., Davis, J. M., and Bloomfield, P.: An Automated Classification Scheme Designed to Better Elucidate the Dependence of Ozone on Meteorology, J.appl.meteor, 33, 1182-1199, 1994.

Han, H., Liu, J., Shu, L., Wang, T., and Yuan, H.: Local and synoptic meteorological influences on daily variability of summertime surface ozone in eastern China, Atmospheric Chemistry and Physics Discussions, 1-51, 10.5194/acp-2019-494, 2019.

Hegarty, J., Mao, H., and Talbot, R.: Synoptic controls on summertime surface ozone in the northeastern United States, Journal of Geophysical Research, 112, 10.1029/2006jd008170, 2007.

Huth, R.: An intercomparison of computer-assisted circulation classification methods, International Journal of Climatology: A Journal of the Royal Meteorological Society, 16, 893-922, 1996.

Huth, R., Beck, C., Philipp, A., Demuzere, M., Ustrnul, Z., Cahynová, M., Kyselý, J., and Tveito, O. E.: Classifications of atmospheric circulation patterns, Annals of the New York Academy of Sciences, 1146, 105-152, 2008.

Jenkinson, A. F., Collison, F.P: An initial climatology of gales over the North Sea. , Synoptic Branch Memorandum No. 62. Met Office, Exeter., 1977.

Jones, P. D., Hulme, M., and Briffa, K. R.: A comparison of Lamb circulation types with an objective classification scheme, International Journal of Climatology, 13, 655-663, 1993.

Lamb, H. H.: British Isles weather types and a register of the daily sequence of circulation patterns, 1861–1971., Geophysical Memoir., 116, p. 85., 1972.

Li, K., Jacob, D. J., Liao, H., Shen, L., Zhang, Q., and Bates, K. H.: Anthropogenic drivers of 2013–2017 trends in summer surface ozone in China, Proceedings of the National Academy of Sciences, 116, 422-427, 10.1073/pnas.1812168116, 2019.

Liao, Z., Gao, M., Sun, J., and Fan, S.: The impact of synoptic circulation on air quality and pollution-related human health in the Yangtze River Delta region, The Science of the total environment, 607-608, 838-846, 10.1016/j.scitotenv.2017.07.031, 2017.

Mckendry, I. G., Stahl, K., and Moore, R. D.: Synoptic sea-level pressure patterns generated by a general circulation model: comparison with types derived from NCEP/NCAR re-analysis and implications for downscaling, International Journal of Climatology, 26, 1727-1736, 2006.

Otero, N., Sillmann, J., Schnell, J. L., Rust, H. W., and Butler, T.: Synoptic and meteorological drivers of extreme ozone concentrations over Europe, Environmental Research Letters, 11, 2016.

Pope, R. J., Butt, E. W., Chipperfield, M. P., Doherty, R. M., Fenech, S., Schmidt, A., Arnold, S. R., and Savage, N. H.: The impact of synoptic weather on UK surface ozone and implications for premature mortality, Environmental Research Letters, 11, 124004, 10.1088/1748-9326/11/12/124004, 2016.

Querol, X., Alastuey, A., Pandolfi, M., Reche, C., Perez, N., Minguillón, M. C., Moreno, T., Viana, M., Escudero, M., and Orio, A.: 2001–2012 trends on air quality in Spain, Science of the Total Environment, 490, 957-969, 2014.

Santurtún, A., González-Hidalgo, J. C., Sanchez-Lorenzo, A., and Zarrabeitia, M. T.: Surface ozone concentration trends and its relationship with weather types in Spain (2001–2010), Atmospheric Environment, 101, 10-22, 2015.

Shen, L., Mickley, L. J., and Tai, A. P. K.: Influence of synoptic patterns on surface ozone variability over the eastern United States from 1980 to 2012, Atmospheric Chemistry and Physics, 15, 10925-10938, 10.5194/acp-15-10925-2015, 2015.

Tang, G., Wang, Y., Li, X., Ji, D., Hsu, S., and Gao, X.: Spatial-temporal variations in surface ozone in Northern China as observed during 2009–2010 and possible implications for future air quality control strategies, Atmospheric Chemistry and Physics, 12, 2757-2776, 10.5194/acp-12-2757-2012, 2012.

Trigo, R. M., and Dacamara, C. C.: Circulation weather types and their influence on the precipitation regime in Portugal, International Journal of Climatology, 20, 1559-1581, 2000.

Yarnal, B.: Synoptic climatology in environmental analysis: a primer, Belhaven, 1993.